# Detection of Seasonal Deformation of Highway Overpasses Using the PS-InSAR Technique: A Case Study in Beijing Urban Area

**Mingyuan Lyu** [1,2,3], **Yinghai Ke** [1,2,3,4,5], **Xiaojuan Li** [1,2,3,4,5,*], **Lin Zhu** [1,2,3,4,5], **Lin Guo** [1,2,3,4,5] **and Huili Gong** [1,2,3,4,5]

1 College of Resources Environment and Tourism, Capital Normal University, 105 North Road of the 3rd Ringroad, Haidian District, Beijing 100048, China; mingyuanlyu@cnu.edu.cn (M.L.); yke@cnu.edu.cn (Y.K.); 5533@cnu.edu.cn (L.Z.); 6382@cnu.edu.cn (L.G.); 4039@cnu.edu.cn (H.G.)
2 Laboratory Cultivation Base of Environment Process and Digital Simulation, Beijing 100048, China
3 Key Laboratory of Mechanism, Prevention and Mitigation of Land Subsidence, MOE, Capital Normal University, Beijing 100048, China
4 Beijing Laboratory of Water Resources Security, Beijing 100048, China
5 Observation and Research Station of Groundwater and Land Subsidence in Beijing-Tianjin-Hebei Plain, MNR, Beijing 100048, China
* Correspondence: lixiaojuan@cnu.edu.cn

**Abstract:** In urban areas, deformation of transportation infrastructures may lead to serious safety accidents. Timely and accurate monitoring of the structural deformation is critical for prevention of transportation accidents and assurance of construction quality, particularly in areas with regional land subsidence, such as the city of Beijing. In this study, we proposed a method for the detection of seasonal deformation of highway overpasses using the integration of persistent scatterers Interferometric Synthetic Aperture Radar (PS-InSAR) techniques and seasonal indices, i.e., deformation concentration degree (DCD) and deformation concentration period (DCP) indices. Taking eastern Beijing urban area as a case study area, we first used the PS-InSAR technique to derive time series surface deformation based on 55 TerraSAR-X images during 2010–2016. Then, we proposed DCD and DCP indices to characterize seasonal deformation of 25 highway overpasses in the study area, with DCD representing to what degree the annual deformation is distributed in a year, and DCP representing the period on which deformation concentrates in the year. Our results showed that the maximum annual deformation rate reached −141.3 mm/year in Beijing urban area, and the PS-InSAR measurements agreed well with levelling measurements ($R^2 > 0.97$). For PS pixels with DCD ≥ 0.3, the monthly deformation showed obvious seasonal patterns with deformation values during some months greater than those during the other months. DCP revealed that the settlement during autumn and winter was more serious than that in spring and summer. The seasonal patterns seemed to be related to the location, structure, and construction age of the overpasses. The upper-level overpasses, the newly constructed overpasses, and those located in the subsidence area (rate < −40 mm/year) tended to show a greater seasonal pattern. The seasonal deformation variations were also affected by groundwater-level fluctuation, temperature, and compressible layer.

**Keywords:** seasonal deformation; overpasses; Beijing urban area; PS-InSAR

## 1. Introduction

Land subsidence is a geological hazard mainly caused by human activities, such as subsurface fluid withdrawal, underground mining, and engineering construction. Land subsidence can induce a

series of geological disasters, such as foundation sinking, house cracking, and underground pipeline damage. Ground fissures resulting from uneven land subsidence can lead to damages to buildings, dams, overpasses, and other urban facilities [1]. As important transportation facilities, deformation of overpasses may cause serious safely issues. Uneven land subsidence may lead to uneven deformation of the highway overpass and partial damage of the bridge [2]. Dewatering and tunneling when constructing metro stations may also cause the pile foundation settlement of overpasses [3].

The overpasses are not only affected by regional land subsidence but also by seasonal deformation of the bridge body. Milillo et al. revealed that the bridge was undergoing an increased magnitude of deformations over time prior to its collapse [4]. Fornaro et al. investigated the thermal response of the Musmeci bridge in Potenza (Italy), and showed that the deformation of bridge was highly correlated with temperature by using the multidimensional imaging (MDI) approach [5]. Lazecky et al. used the PS-InSAR to demonstrate the bridge deformation in Bratislava, Ostrava, and Hong Kong due to thermal dilation of the structure [6]. Crosetto et al. measured thermal expansion of a viaduct by adopting the PS-InSAR technique, and found the thermal expansion range between the two subsequent joints was $-0.35\sim0.3$ mm/°C [7]. Zhao et al. found the seasonal deformation pattern for several PS points in Lupu Bridge in Shanghai and analyzed the relationship with temperature [8]. Deformation of highway overpass bridges may cause disasters, such as cracks and collapses of bridges, which endangers people's property and life safety. Understanding the characteristics of overpass deformation is of great importance for the prevention and control of overpass damages. This is especially important for cities with regional land subsidence, as the background deformation may aggravate the deformation of highway overpasses.

Land subsidence has threatened many countries, including China. Beijing, the capital city of China, has been severely affected by land subsidence since 1990s. By the end of 2013, the area with a total settlement of more than 50 mm had reached more than 4200 km$^2$, accounting for 66% of the total area of Beijing plain [9]. Previous studies showed that land subsidence in the urban area is serious [10,11]. The largest settlement center is located in the east of Chaoyang district [12–15]. At the same time, Beijing has the largest number of highway overpasses in China. Previous research has mostly focused on regional land subsidence monitoring in Beijing [16–18]. Few studies have been conducted on characterizing deformation of transportation lines.

Conventional deformation monitoring techniques include levelling, Global Positioning System (GPS) measurements, stratified markers, borehole extensometers, etc [19–21]. However, these approaches have the drawbacks of a low spatial sampling density, long observation period, and high cost [22]. Compared to the conventional methods, persistent scatterers Interferometric Synthetic Aperture Radar (PS-InSAR) has the capability to obtain large-scale surface deformation over wide area [23], and has been widely used for regional land deformation monitoring [9,10,24–26]. With the development of SAR technology, SAR-based monitoring has become valuable for monitoring the deformation of infrastructure elements, such as bridge displacement, roadway surface deformation, etc. [27]. It also has the potential of structural damage assessment [28].

However, previous studies usually relied on the annual deformation rate or cumulative settlement to characterize ground deformation patterns. The intra-annual variations of deformation have not been fully investigated. Zhang et al. reported that the deformation of the aquitards has the characteristics of elastic deformation within a year based on levelling and borehole-extensometer data [20]. Hu et al. analyzed the spatial-temporal distribution of land subsidence in Beijing by the small baseline subset (SBAS) technique, and found the deformation of five feature points has seasonal fluctuation [29]. Zhu et al. analyzed the deformation recorded at three depth intervals of an extensometric station, and found the seasonal behavior of the piezometric head caused a slight seasonal deformation of the corresponding layers [9]. It can be seen that current studies on seasonal characteristics of deformation are primarily based on levelling measurements and several persistent scatterer (PS) pixels, which cannot quantitatively represent the whole study area.

To date, several models and indices have been developed to analyze seasonality in time series data. Typical methods include decomposing time series into level, trend, seasonality, and noise using additive or multiplicative models, or using Loess or STL decomposition [30]. These methods are suitable for long-term series data that are periodic and are frequently used for time series forecasting. Another type of method was to use indices to measure the distribution and concentration of time series variables within a certain period, and have been widely used to analyze the intra-annual pattern of climatological or hydrological variables, such as precipitation and streamflow. Such indices include the modified Fournier index (MFI), precipitation concentration index (PCI), precipitation concentration index (PCD), precipitation concentration period (PCP), etc. [31]. Zhao et al. pointed out that PCD and PCP [32] were more suitable for representing the concentration, barycentre date of precipitation, and are promising for characterizing intra-annual variations of other time series variables.

In this study, we aimed to (1) present a method to quantify seasonal deformation using time-series displacement derived from the PS-InSAR technique, and (2) analyze the characteristics of seasonal deformation of overpasses by choosing eastern Beijing urban area as the study area. First, we derived time series land surface deformation in eastern Beijing urban area during 2010–2016 using the PS-InSAR method based on 55 TerraSAR-X images. Then, based on the time-series deformation on the highway overpasses, we proposed the deformation concentration degree (DCD) and deformation concentration period (DCP) to characterize seasonal deformation. Finally, we analyzed the characteristics and the causes of seasonal deformation on the overpasses.

In this manuscript, the study area, datasets, and properties of the overpasses are summarized in Section 2. Then, the PS-InSAR method and new parameters (DCD and DCP) used in this study are presented in Section 3. A detailed description of the subsidence rate and seasonal characteristics on overpasses from the PS-InSAR technique and new parameters (DCD and DCP) are seen in Section 4. Finally, the discussion of the causes of seasonal deformation is given in Section 5, and the main conclusions are summarized in Section 6.

## 2. Study Area and Data Sets

### 2.1. Study Area

Beijing (115°25′–117°30′E, 39°28′–41°05′ N) is located in the northern part of the North China Plain, with a total area of 16,422.78 km$^2$. The region can be divided into the western mountain area, the northern mountain area, and the southeast plain area in elevation from northwest to southeast. Climatically, Beijing has a monsoon-influenced semi-arid and semi-humid continental climate with high temperatures in the summer and low temperatures in the winter, and abundant precipitation. The temporal and spatial distribution of annual precipitation in Beijing area is uneven. The average annual precipitation is approximately 583 mm, concentrated in the summer season from June to September. The study area is located in the eastern Beijing urban area (mostly in Chaoyang district), with an area of 403.81 km$^2$ (Figure 1). So far, the study area contains two major ground subsidence funnels, with the funnel in Chaoyang District having the longest subsidence history in the Beijing plain [10,25].

### 2.2. Datasets

Due to a shorter wavelength and better spatial and temporal resolution, current X-band SAR systems are more sensitive to thermal expansion [6]. This type of data provides more details about individual objects and a higher density of PS points [33]. Compared with other moderate-resolution satellite sensors, such as ERS1/2, Envisat, and PALSAR, the high-resolution TerraSAR-X image is very advantageous in that it has a remarkable improvement of spatial resolution and thus more PS pixels can be extracted and the PS density can be boosted in urban districts and suburbs [34]. In this study, TerraSAR-X images were used to monitor land subsidence (as shown in Figure 1). The SAR datasets used in this study include 55 images collected by X-band TerraSAR-X images with HH polarization

acquired along ascending orbits (track 8) collected from 13 April 2010 to 24 May 2016. The operational mode of TerraSAR-X is Stripmap mode. The spatial resolution of TerraSAR-X is 3 m on the ground. Detailed information of the SAR datasets is summarized in Table 1. The SRTM-3 (Shuttle Radar Topography Mission) 90-m digital elevation data were downloaded from the United States Geological Survey (http://dds.cr.usgs.gov/srtm/) and used to remove the topographic phase contribution in the processing of the multi-temporal InSAR method.

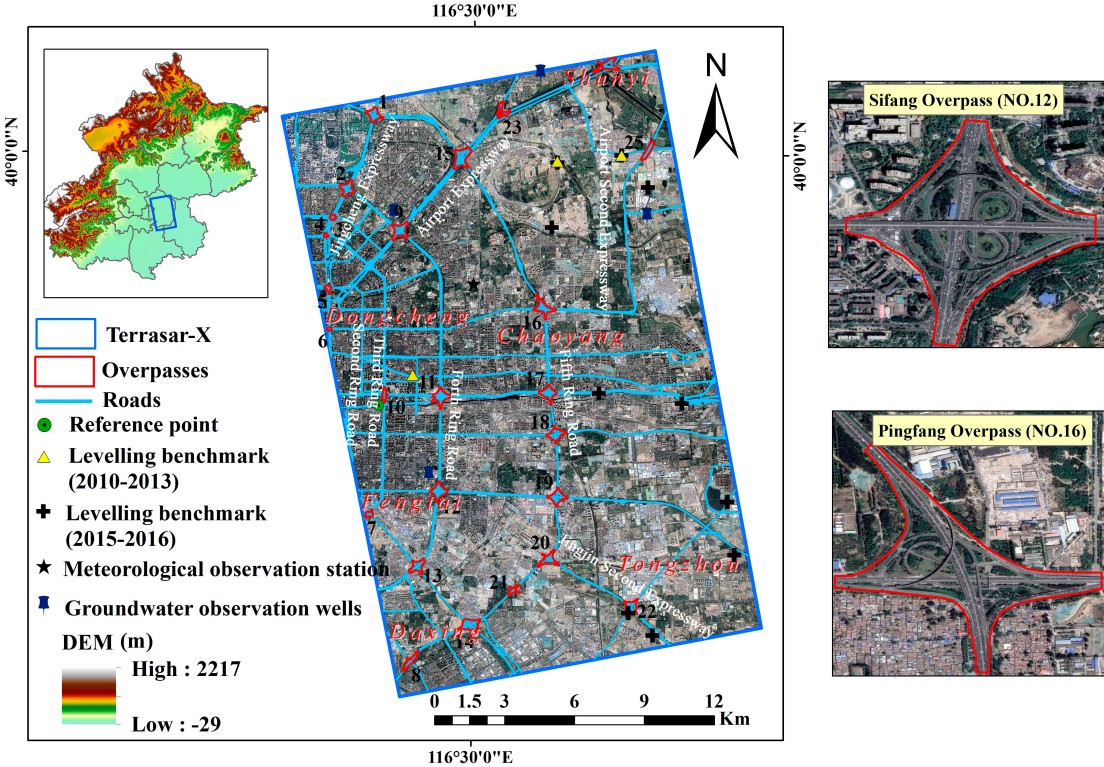

**Figure 1.** Location of the study area and coverage of the datasets.

**Table 1.** Properties of the TerraSAR-X dataset.

| Parameter | TerraSAR-X |
|---|---|
| Band | X |
| Wavelength (cm) | 3.1 |
| Incidence angle (°) | 33.1 |
| Heading (°) | −10 |
| Polarization | HH |
| Spatial resolution (m) | 3 |
| Track No. | 8 |
| Orbit direction | Ascending |
| No. of image | 55 |
| Data range | 13 April 2010–24 May 2016 |

Land surface deformation measurements collected from 13 levelling benchmarks, including 3 levelling benchmarks from 2010–2013 and 2015–2016, and 10 levelling benchmarks from 2015–2016, were used for validation (Figure 1). Due to the limited availability of long-term observations, none of the in situ datasets covered the whole time span from 2010 to 2016.

*2.3. Properties of Overpasses*

　　By 1996, there have been more than 160 highway overpasses in Beijing, ranking the first in China in terms of total number, individual dimensions, and the variety of styles of highway overpasses [35]. By far, Beijing has more than 700 highway overpasses, accounting for more than 70% of the total number of overpasses in China. The overpasses have various types with complex structures, including cloverleaf, diamond, rotary, trumpet, directional, and combination type, and the most common one is a diamond overpass. In the study area, there are a total of 25 large highway overpasses with complex structures (Figure 1). The boundaries of these overpasses were manually outlined based on Google Earth images. These polygons covered overpasses and the adjacent roads. Figure 1 demonstrates the Google Earth imagery over two cloverleaf overpasses, Sifang Overpass (NO.12) and Pingfang Overpass (NO.16), respectively. As shown in Figure 1, the 25 overpasses are distributed on major trunk roads and ring roads. The properties of 25 overpasses are shown in Table 2. Dongzhimen North Overpass (NO.5) located on the Second Ring Road is the oldest overpass, which was constructed in 1980.

**Table 2.** Properties of the 25 overpasses in the study area.

| Overpasses Number | Name | Location | Age of Construction |
|---|---|---|---|
| 1 | Laiguangying Overpass | Fifth Ring Road | 2002 |
| 2 | Wanghe Overpass | Fourth Ring Road | 2002 |
| 3 | Shaoyaoju Overpass | JingCheng Expressway | 2002 |
| 4 | Taiyanggong Overpass | Third Ring Road | 1994 |
| 5 | Dongzhimen North Overpass | Second Ring Road | 1980 |
| 6 | Dongsishitiao Overpass | Second Ring Road | 1982 |
| 7 | Fenzhongsi Overpass | Third Ring Road | 1990 |
| 8 | Yizhuang Overpass | Fifth Ring Road | 2003 |
| 9 | Siyuan Overpass | Fourth Ring Road | 1993 |
| 10 | East Third Ring Road Overpass | Third Ring Road | 1984 |
| 11 | Sihui Overpass | Fourth Ring Road | 1995 |
| 12 | Sifang Overpass | Fourth Ring Road | 1999 |
| 13 | Shibalidian Overpass | Fourth Ring Road | 1990 |
| 14 | Dayangfang Overpass | Fifth Ring Road | 1990 |
| 15 | Wuyuan Overpass | Fifth Ring Road | 2003 |
| 16 | Pingfang Overpass | Fifth Ring Road | 2003 |
| 17 | Yuntong Overpass | Fifth Ring Road | 2003 |
| 18 | Xiaojiaoting Overpass | Fifth Ring Road | 2003 |
| 19 | Wufang Overpass | Fifth Ring Road | 2003 |
| 20 | Huagong Overpass | Fifth Ring Road | 2003 |
| 21 | Kanghua Overpass | Fifth Ring Road | 2003 |
| 22 | Dongshicun Overpass | Jingjin Expressway | 2008 |
| 23 | Beipu Overpass | Airport Expressway | 2008 |
| 24 | Wenyu Overpass | Airport Expressway | 2008 |
| 25 | Jinzhan Overpass | Airport Second Expressway | 2002 |

## 3. Methods

　　First, we employed the PS-InSAR method in SARPROZ software to derive the annual displacement rate and cumulative displacement based on TerraSAR-X images over the urban area of Eastern Beijing. Second, we presented two seasonal indices, i.e., displacement concentration degree (DCD) and displacement concentration period (DCP) based on the monthly average displacement of each PS pixel, to quantify the characteristics of seasonal deformation of the overpasses. Finally, according to the spatial and temporal difference distribution of DCD and DCP in the study area, the seasonal characteristics of overpass deformation and the causes are analyzed and discussed (Figure 2).

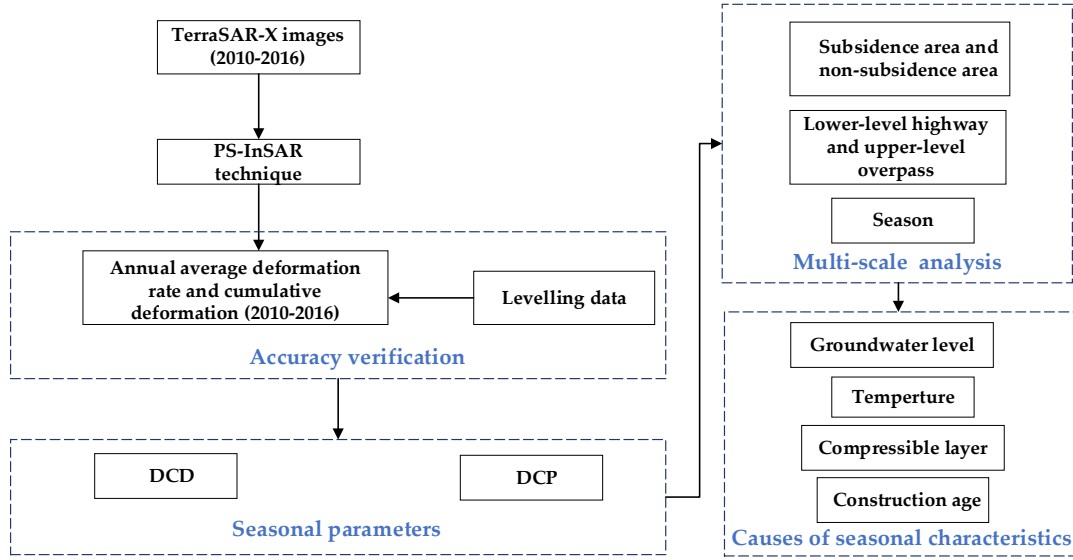

**Figure 2.** The workflow for monitoring seasonal deformation of highway overpasses.

### 3.1. PS-InSAR

In this study, the PS-InSAR method in SARPROZ software was used to derive time-series deformation based on TerraSAR-X images over the study area. The PS-InSAR approach was first presented by Ferretti et al. [23]. Although various algorithms of PS-InSAR techniques have been developed in recent years, all these algorithms aim to retrieve local deformation over highly coherent scatterers, namely persistent scatterer (PS) pixels, such as buildings or bare rock objects, from the wrapped differential interferometric phase.

The main formula of PS-InSAR method is as follows:

$$\phi = W\{\phi_{def} + \phi_{atm} + \phi_{orb} + \phi_{\varepsilon} + \phi_{noise}\}, \tag{1}$$

where $W\{\ \}$ is the wrapping operator, and $\phi_{def}$ represents the phase change due to ground motion in the line-of-sight (LOS) direction between two satellite passes, which is related in the time domain and spatial domain. $\phi_{atm}$ denotes the phase due to the difference in atmospheric delay, which is uncorrelated in the time domain and correlated in the spatial domain. $\phi_{orb}$ and $\phi_{\varepsilon}$ are the residual phase due to inaccurate estimation of orbital parameters and DEM errors, respectively. $\phi_{noise}$ is the noise phase, including scattering, co-registration errors, thermal noise, and uncertainty, in the position of the phase center at the azimuth. Among these five components, $\phi_{def}$ is expected to be accurately obtained by removing or minimizing other error components' effects.

By using PS-InSAR technology in SARPROZ software (refer to https://www.sarproz.com/), deformation time series were obtained from TerraSAR-X. The TerraSAR-X image acquired on 1 November 2013 was selected as the master image, and other images were co-registered to the master image. Figure 3 shows the baseline information of TerraSAR-X images. The center of the connecting lines was the master scene. The topographic phase contribution was removed using SRTM DEM (spatial resolution of 90 m). Then, persistent scatterer candidates (PSCs) were obtained with an amplitude difference dispersion index lower than 0.3. Multi-image grid phase unwrapping was then conducted, and an atmospheric phase screen (APS) was estimated and removed based on a given reference point. The location of the reference point is shown in Figure 1. The distance of between the reference point and the nearest overpass (East Third Ring Road Overpass (No.10)) was 225 m. Afterwards, PS points with a temporal coherence index greater than 0.75 were selected. Finally, the time

series of the deformation along the line-of-sight (LOS) was derived by SARPROZ software. The vertical deformation rate was then estimated from the LOS deformation [36]:

$$v_{u-d}(x,y) = v_{LOS}(x,y) / \cos\theta_{(x,y)}, \tag{2}$$

where $v_{u-d}(x,y)$ is the up-down (vertical) deformation rate, $v_{LOS}(x,y)$ represents the SAR LOS deformation rate, and $\theta_{(x,y)}$ denotes the SAR viewing angle at position $(x,y)$.

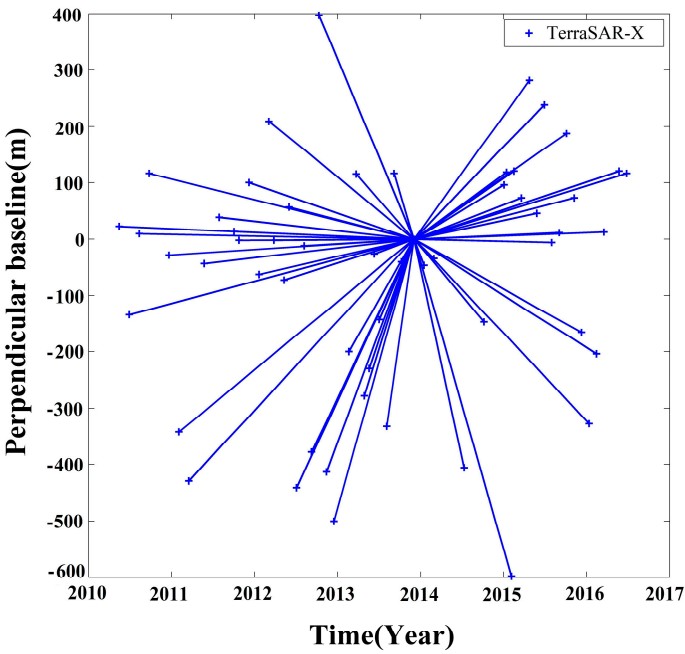

**Figure 3.** Baseline information for the TerraSAR-X datasets.

### 3.2. Characterizing Seasonal Deformation Using DCD and DCP

The PS-InSAR technique provided the annual average deformation rate and cumulative deformation of each PS point, which cannot represent the intra-annual variations of deformation. Here, we introduced the concentration degree (CD) and concentration period (CP), which are frequently used to characterize concentration characteristics of intra-annual distribution of hydrological variables, for seasonal deformation detection. CD and CP were first proposed by Zhang et al. [32]. They were first used in precipitation and were also used to analyze streamflow [32,37–40]. Higher concentrations of precipitation or streamflow are represented by higher percentages of the yearly precipitation or streamflow in a certain period. In this study, we adopted CD and CP, and proposed the deformation concentration degree (DCD) and deformation concentration period (DCP) as new indictors to represent the intra-annual and seasonal variations of deformation. DCP represents the period (months) during which the total deformation is concentrated, and DCD represents to what degree the deformation is distributed across 12 months.

The basic principle of DCD and DCP was based on the vector composition, considering monthly deformation as a vector. First, for each PS pixel, deformation in each month from 2010 to 2016 was obtained from cumulative displacement using the linear temporal interpolation method. The monthly average deformation was then calculated by averaging the monthly deformation across the 7 years. The deformation can be divided into settlement (vertical displacement value is negative) and uplift (vertical displacement value is positive). For seasonal index calculation, all values need to be positive; thus, we converted negative displacement values to positive values by adding the original monthly deformation to the absolute value of minimum deformation (Equation (3)):

$$r_i = D_i + \left|\min(D_i,\ i = 1, 2, \ldots 12)\right|, \tag{3}$$

where $D_i$ denotes the monthly average deformation value in month $i$ ($i = 1, 2, \ldots 12$), and $r_i$ denotes the adjusted deformation value.

$r_i$ was then taken as the length of the vector, and the corresponding month was considered as the vector direction. The whole year was considered as a circle (360°), and then one month corresponded to 360°/12 = 30° (Figure 4). The DCD and DCP can be calculated as follows:

$$R_x = \sum_{i=1}^{12} r_i \sin \theta_i, \tag{4}$$

$$R_y = \sum_{i=1}^{12} r_i \cos \theta_i, \tag{5}$$

$$R = \sqrt{R_x^2 + R_y^2}, \tag{6}$$

$$DCD = \frac{R}{\sum_{i=1}^{12} r_i}, \tag{7}$$

$$DCP = \arctan(\frac{R_x}{R_y}), \tag{8}$$

where $\theta_i$ stands for the angle value of the $i$th month. $R_x$ and $R_y$ represent the horizontal and vertical components of the deformation, respectively, and $R$ represents the annual total deformation in a year. DCD represents the degree of $R$ concentrated among 12 months, ranging from 0 and 1. When $r_i$ is concentrated in a single month, i.e., $D_i$ is high for that month but equally low in other months, the DCD reaches its maximum value. When the deformation is evenly distributed across 12 months, the DCD reaches its minimum value. DCP represents the month when the total $R$ is concentrated in. Taking a PS pixel as an example (Figure 4), the vector resultant of deformation in a year is $R$, and its position is shown in the Figure 4. DCP of this PS pixel is 140° and the corresponding month is May, indicating that the vertical displacement reaches its maximum value in May. In other words, the settlement in May was the smallest.

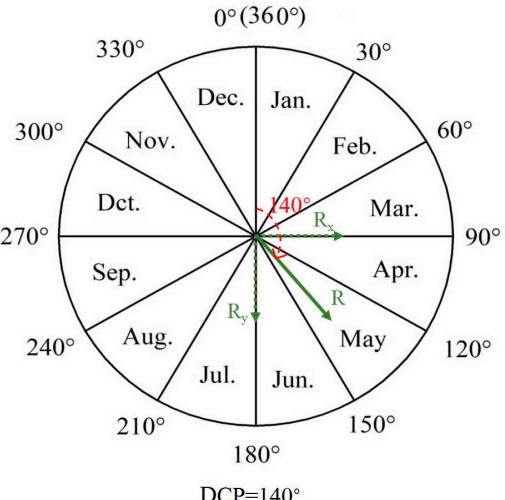

**Figure 4.** Illustration of DCP calculation and the meaning.

For each PS pixel, DCD and DCP, representing the seasonal characteristics of deformation, were obtained respectively. The spatial and temporal differences of seasonal deformation on the overpasses were analyzed on multiple scales. Finally, the causes of seasonal deformation difference of overpass were discussed (Figure 2).

## 4. Results

### 4.1. The Results of PS-InSAR and Validation

Figure 5 illustrates the resultant annual deformation rate in the eastern Beijing urban area. A total of 345,550 PS pixels were identified from the TerraSAR-X. It can be seen that the uneven settlement was obvious, and the surface displacement rates vary widely throughout the study area. The rate ranged from −141.3 to 15 mm/year during 2010–2016. Moreover, two subsidence funnels formed, including Dongbalizhuang-Dajiaoting (DD) and Chaoyang Jinzhan (CJ) (Figure 5). The maximum displacement rates were −136.87 and −141.3 mm/year in the DD and CJ subsidence funnel, respectively.

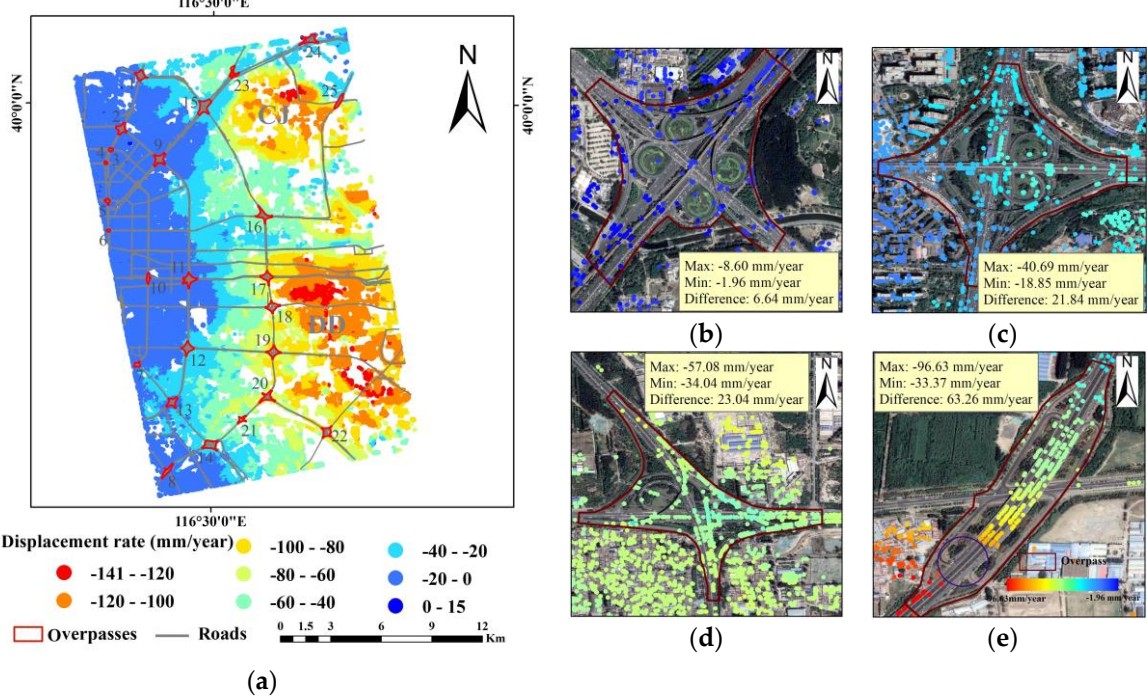

**Figure 5.** (**a**) Displacement rates from 2010 to 2016 in the study area; (**b–e**) The difference of the displacement rate on Siyuan overpass (NO.9), Sifang overpass (NO.12), Pingfang overpass (NO.16), and Jinzhan overpass (NO.25).

There were a total of 4599 PS points over the overpass polygons, meaning that an average of 184 PS points were located over each overpass. The maximum displacement rate over these overpass polygons was −99.46 mm/year, which was located on Dongshicun Overpass (NO.22). In our study area, the mean deformation of overpasses in the east was more serious than that in the west. This is consistent with the overall spatial distribution of land subsidence in the study area reported in previous research [15,24]. Siyuan Overpass (NO.9) has a low deformation rate (−5.18 mm/year), and the difference of the deformation rate within the overpass polygon was also low (6.64 mm/year). Among the four overpasses, Jinzhan overpass (NO.25) has the greatest deformation rate (−56.19 mm/year), and the difference of the deformation rate within the overpass polygon was highest (63.26 mm/year). The reason for the difference is that Jinzhan overpass (NO.25) passes through the CJ settlement funnel zone. In addition, there is no PS point in the purple circle on Jinzhan overpass (NO.25) (Figure 5e); because of the limitation of PS-InSAR technology, PS points on the flat road with low temporal coherence are less.

The mean deformation rate during the two time periods was further assessed by 3 in situ levelling measurements collected from 2010 to 2013 and 10 levelling measurements collected from 2015 to 2016 (Figure 1). As illustrated in Figure 6, the InSAR measurements show good consistency with the levelling

measurements. Both measurements are strongly correlated, and the coefficient of determination ($R^2$) of the linear regression between them are 0.99 and 0.97. The maximum absolute errors are 8.9 and 9.86 mm/year, and the minimum absolute errors are 0.88 and 4.28 mm/year. The root-mean-square-errors (RMSE) are 7.54 and 4.18 mm/year, indicating the reliability of the InSAR results. Figure 6b,c compares the time series cumulative deformation from levelling measurements and that from PS-InSAR at two levelling benchmarks. It is observed that the general trend of land subsidence derived from PS-InSAR agrees very well with the levelling measurements. It indicates that the InSAR results can meet the requirements of ground deformation monitoring accuracy.

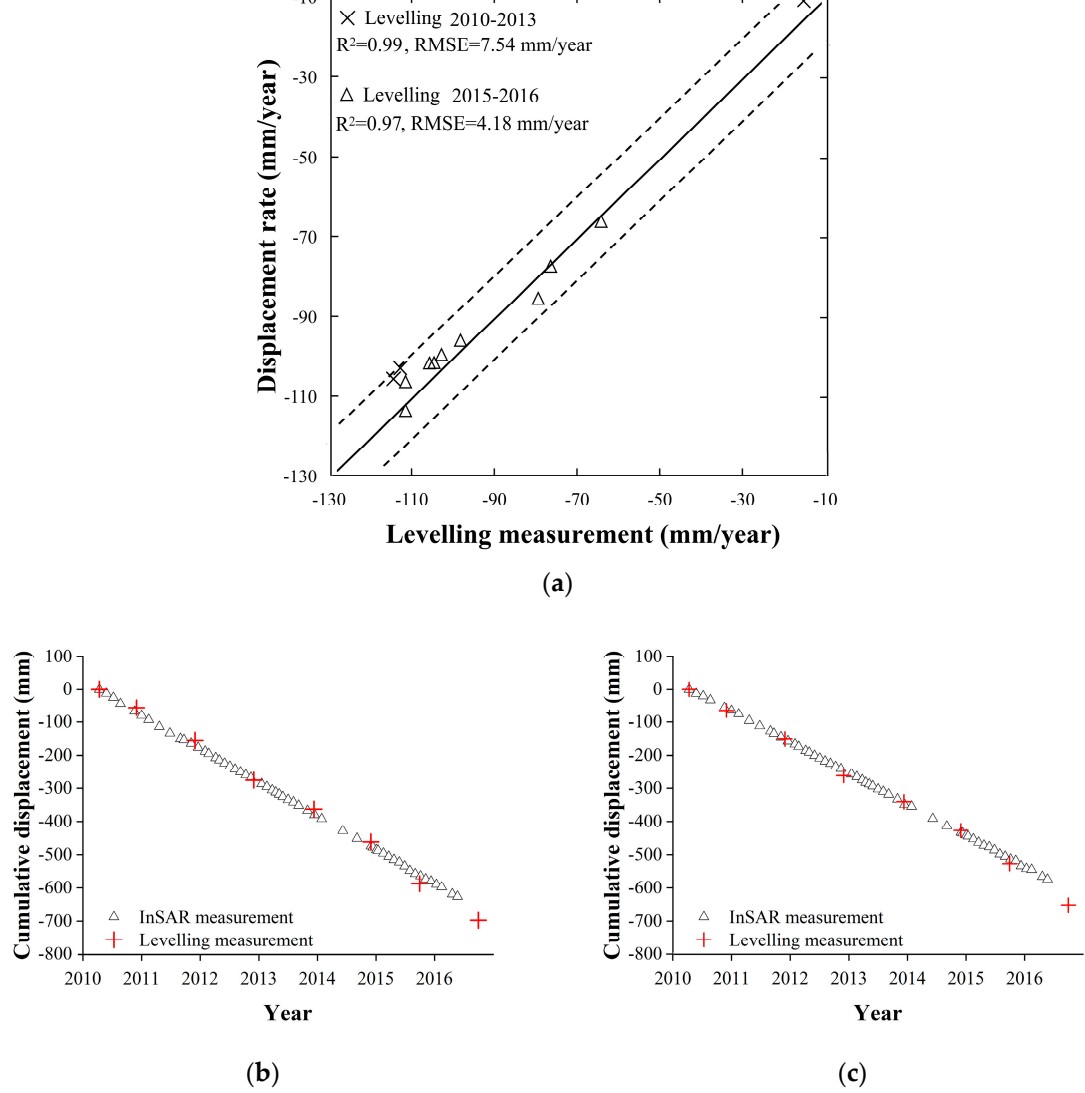

**Figure 6.** Comparison of the InSAR results and levelling measurement. (**a**) Comparison of the annual displacement rates of InSAR results and levelling measurement. (**b**,**c**) Time series cumulative displacement from the InSAR results and levelling measurement.

### 4.2. Seasonal Characteristics of Deformation on Overpasses

Using the method in Section 3.2, DCD was calculated for each PS pixel. Among the PS pixels within the overpass polygons, the maximum value of DCD was 0.65. Figure 7 shows the monthly deformation of the PS pixels with DCD values ranging from 0.01 to 0.65 (0.01, 0.20, 0.25, 0.30, 0.50, 0.65, and 0.65). It can be seen that for the pixels with a DCD value greater than 0.3, the vertical displacement values during several continuous months were higher than that during other months. For example,

for a pixel with DCD = 0.65, vertical displacement during March–June was around −4.4 mm/month, while during August–December, the displacement value was around −5.4 mm/month. Therefore, 0.3 was selected as the threshold that determined whether the PS pixel has a seasonal pattern. PS pixels with DCD values over 0.3 were considered as those that have seasonal patterns, meaning that a high value of vertical displacement concentrated in a few months, and the concentration degree of the total deformation during 12 months was relatively high. As shown in Figure 7d, the peak of deformation occurs in spring, and the valley of deformation usually occurs in autumn. Figure 7e–g display the change tendencies similar to Figure 7d. From Figure 7h, the distribution of settlement is strongly seasonal during 2010 to 2016. It demonstrates that using DCD to indicate the seasonal distribution of deformation is reliable.

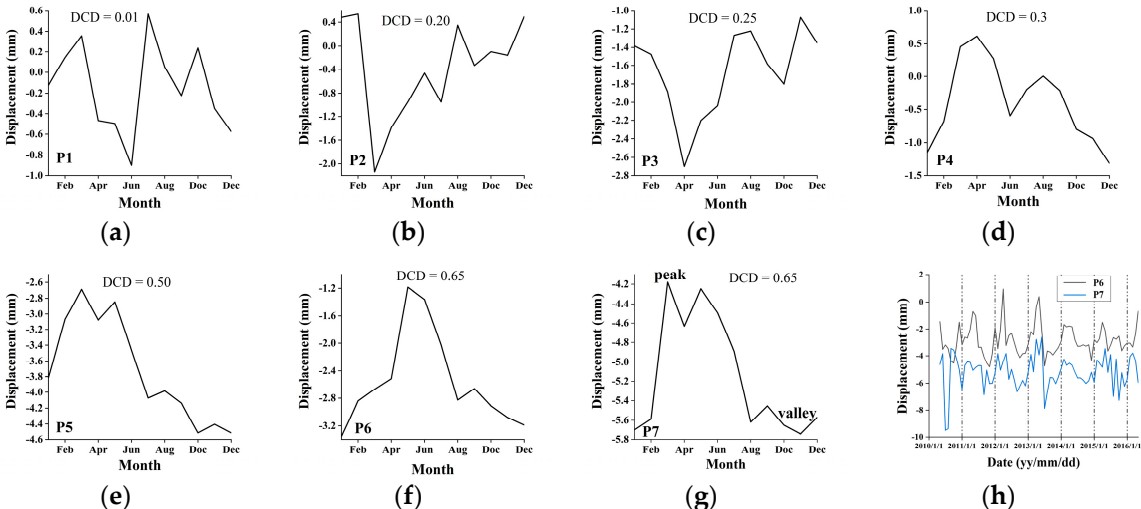

**Figure 7.** The annual average monthly displacement distribution with different DCD. The annual average monthly displacement of (**a**) point 1, (**b**) point 2, (**c**) point 3, (**d**) point 4, (**e**) point 5, (**f**) point 6, and (**g**) point 7; (**h**) The monthly displacement of point 6 and point 7 during 2010 to 2016.

Using 0.3 as the threshold, PS pixels on overpasses were classified as "seasonal" pixels (DCD ≥ 0.3) or "non-seasonal" pixels (DCD < 0.3). By performing Mann–Whitney U tests [41] on DCP of "seasonal" pixels and "non-seasonal" pixels, the selected threshold of DCD is reasonable ($p < 0.05$ for all points on overpasses).

In this study, the area with a settlement rate greater than 40 mm/year is considered as a subsidence area and the remaining area is considered as a non-subsidence area (Figure 8). In terms of spatial distribution, the percentage of overpasses with obvious seasonality is greater in the north than that in the south and greater in the east than in the west. It is consistent with the spatial distribution of land subsidence. There are 15 overpass polygons in the non-subsidence area and 10 overpass polygons in the subsidence area (Table 3). In the non-subsidence area, there are 4 of 15 overpass polygons (26.67%) within which more than 40% of the total amount of PS pixels show seasonal patterns. In the subsidence area, there are 8 of 10 overpasses (80%) with more than 40% of the PS pixels showing seasonal patterns. There are 3 and 5 overpasses having seasonal PS points more than 50% of the total amount of PS pixels in the non-subsidence area and subsidence area, respectively. Compared to the non-subsidence area, the subsidence area had a higher percent of overpasses that showed seasonal patterns. When examining the location of these overpasses, it can be seen that most of those overpasses with seasonal patterns were located in the north, which was consistent with the spatial distribution of land subsidence.

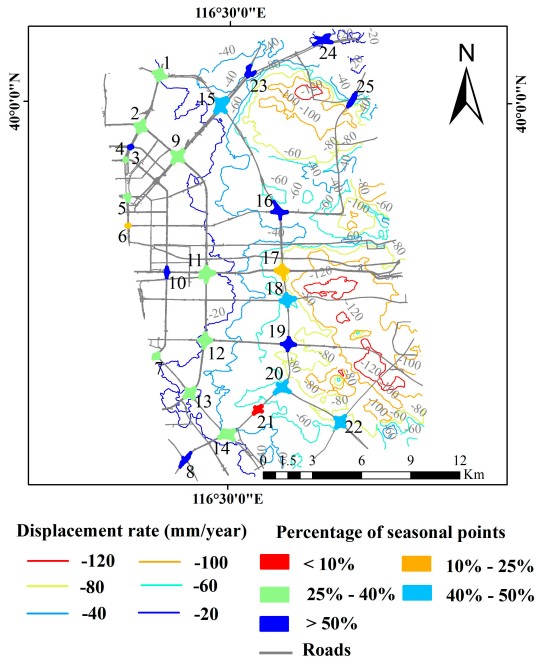

**Figure 8.** The spatial distribution of overpasses, which have different percentages of seasonal points.

**Table 3.** Proportion of seasonal PS points on overpasses in the non-subsidence area and subsidence area.

| Percentage of Seasonal Points | The Number of Overpasses in Non-Subsidence Area (Percentage) | The Number of Overpasses in Subsidence Area (Percentage) |
|---|---|---|
| <10% | 0 | 1 (10%) |
| 10%–25% | 1 (6.67%) | 1 (10%) |
| 25%–40% | 10 (66.66%) | 0 |
| 40%–50% | 1 (6.67%) | 3 (30%) |
| >50% | 3 (20%) | 5 (50%) |

Figure 9 shows the distribution of seasonal PS points and non-seasonal PS points on Siyuan overpass (NO.9), Sifang overpass (No.12), Pingfang overpass (NO.16), and Jinzhan overpass (NO.25). Siyuan overpass (NO.9) and Sifang overpass (No.12) are located in the non-subsidence area, and on the overpasses, 39% and 34% of the PS points are seasonal points. Pingfang overpass (NO.16) and Jinzhan overpass (NO.25) are located in the settlement area. The proportion of seasonal PS points is 51% and 53%, respectively, and the distribution of seasonal PS points is concentrated. Compared with surrounding buildings, the seasonal PS points on the overpasses are denser. It indicates that the deformation of the overpass is more seasonal than other surrounding buildings.

The expansion mode of Beijing urban area is the single-center expansion mode, which takes the central city as the core and expands to the surrounding areas [42,43]. For the Ring Road, the construction years from the Second Ring Road to the Fifth Ring Road are becoming newer. There are 6 overpass polygons on Fifth Ring Road having more than 40% seasonal PS pixels, while only one overpass polygon on Third Ring Road had seasonal points of more than 40% (East Third Ring Road Overpass (NO.10)). None of the overpasses on the Fourth Ring road are seasonal. It indicates that the overpasses in the subsidence area have more obvious seasonal deformation than those in the non-subsidence area, and the newly-constructed overpasses have more remarkable seasonal deformation than the older overpasses.

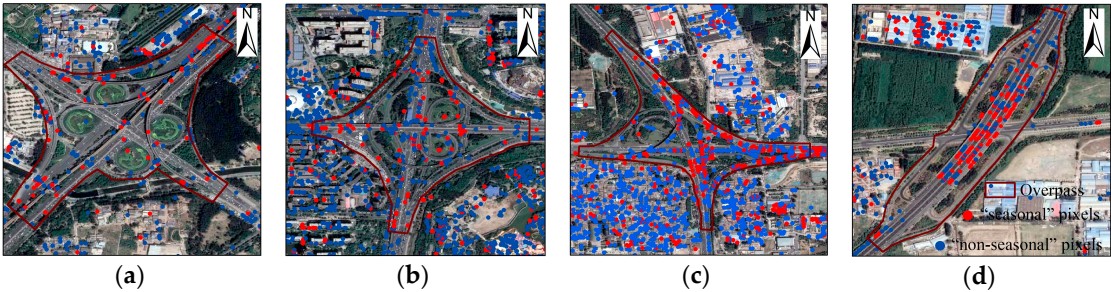

**Figure 9.** The distribution of seasonal PS points and non-seasonal PS points on (**a**) Siyuan overpass (NO.9), (**b**) Sifang overpass (No.12), (**c**) Pingfang overpass (NO.16), and (**d**) Jinzhan overpass (NO.25).

Based on Google earth image and field investigation, the lower-level highway and upper-level overpass are outlined from each overpass polygons by manual delineation. Taking Pingfang overpass (NO.16) as an example, as shown in Figure 10a, the number of seasonal points of the upper-level overpass is larger and the distribution is more concentrated. Figure 10b,c are the local photos of the upper-level overpass and lower-level highway. Figure 11 shows the proportion of seasonal PS pixels at the lower-level highway and the upper-level overpasses. For most of the overpasses (except Dayangfang overpass (NO.14)), the upper-level overpasses have a higher proportion of seasonal PS pixels than the lower-level highway. For the Yuntong overpass (NO.17), the proportion of seasonal points on the upper-level overpass is 40%, while that on the lower-level highway is only 4%. For all overpass polygons, the average proportion of seasonal points in the upper-level overpasses is higher than those in the lower-level highway. It indicates the seasonality is more obvious on the upper-level overpass than lower-level highway.

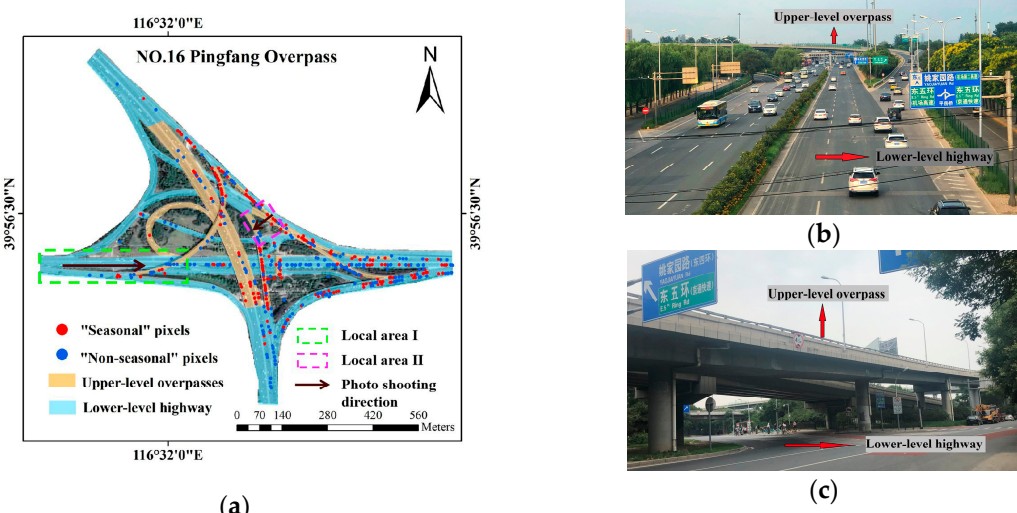

**Figure 10.** (**a**) The PS pixels on upper-level overpass and lower-level highway on Pingfang overpass (NO.16); The photo of the upper-level overpass and lower-level highway (**b**) I and (**c**) II.

DCP describes the period during which the high values of vertical displacement concentrate in the year. Figure 12 illustrates the proportions of seasonal PS pixels with DCP of a given month. As shown in Figure 12a,b, DCP is mainly within the months from March to July for the seasonal pixels both within overpass polygons and on the upper-level overpasses, accounting for 78.47% and 75.65%. As the study area was dominated with settling land surface, this means that the settlements of most seasonal PS pixels had minimum settlement during these months, which is consistent with the temporal variations of deformation shown in Figure 7d–g.

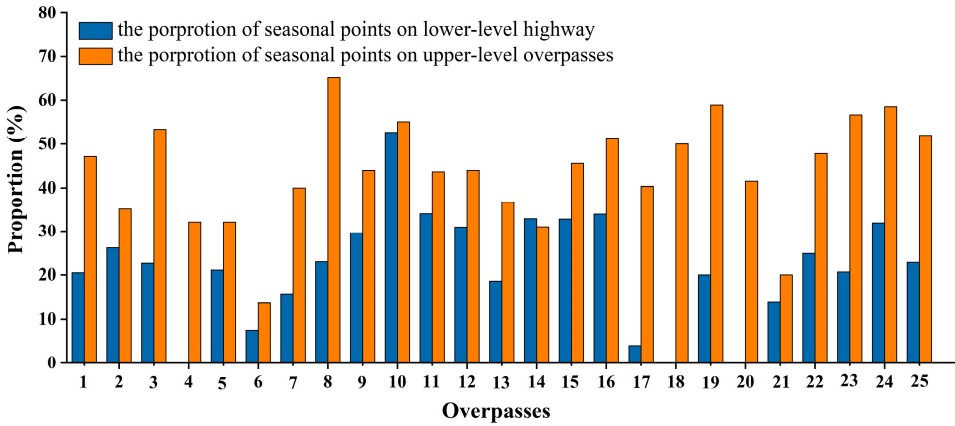

**Figure 11.** The proportion of seasonal points of the upper-level overpasses and lower-level highway.

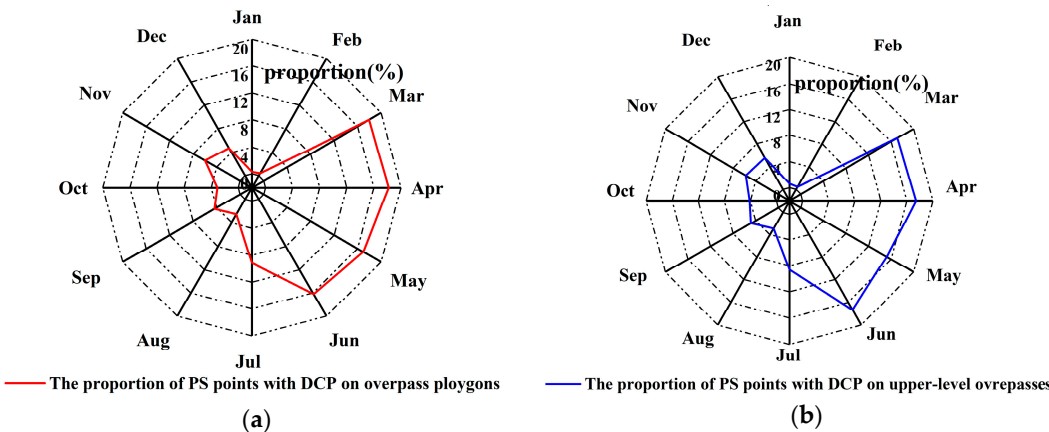

**Figure 12.** The proportion of PS points with a DCP value equal to a given month from January to December. (**a**) PS pixels located within overpass polygons; (**b**) PS pixels located on the upper-level overpasses.

For each of the seasonal PS pixels over the overpasses, we further examined the seasons when the maximum and minimum deformation occurred. Here, we call maximum deformation the peak and minimum deformation the valley. As shown in Table 4, around 65% of the seasonal PS pixels had the valley concentrated in autumn (September–November) and winter (December–February), while 81.1% of the PS pixels had the peaks concentrated in spring (March–May) and summer (June–August). It indicates that the settlement in autumn and winter is more serious than that in spring and summer. In a year, more attention should be paid to the settlement in autumn and winter.

**Table 4.** Proportion of seasonal PS points on overpasses.

| Type | Season | Proportion |
|------|--------|------------|
| Valley | Spring | 13.82% |
| | Summer | 21.43% |
| | Autumn | 37.41% |
| | Winter | 27.34% |
| Peak | Spring | 53.15% |
| | Summer | 27.95% |
| | Autumn | 13.45% |
| | Winter | 5.45% |

## 5. Discussion

### 5.1. The Relationship with Groundwater Level

Previous studies indicated that the main cause of regional land subsidence in Beijing is overexploitation of groundwater [25]. Moreover, the deformation and confined water level show a good correlation [20,26]. The sand initially exhibits elastic deformation and then exhibits plastic deformation with further withdrawal recharge cycles [44]. The confined water level was obtained from groundwater observation wells (Figure 1). Figure 13a–h show the relationship between the mean monthly confined water level measured at four observation wells and the mean monthly deformation of seasonal PS points on the four nearest overpasses (Siyuan overpass (NO.9), Sifang overpass (No.12), Pingfang overpass (NO.16), and Beipu overpass (NO.23)) from 2010 to 2016. As shown in Figure 13c,e,g, the confined water level was high in January and February and was low in June–July. July to September is the rainy season in Beijing, and the groundwater level rises gradually. Cao et al. found that a lagging effect existed between land subsidence and groundwater extraction volume using the polynomial distribution lag (PDL) model [45]. Gao et al. showed the subsidence time series reflect obvious elastic deformation characteristics (seasonal characteristics) as the groundwater-level changes in Eastern Beijing Plain [36]. In addition, the land subsidence time series have several months behind the groundwater-level change. Zhou et al. found a lag period between land subsidence and groundwater-level changes of approximately two or three months in Beijing [46]. Figure 13b,d,f,h show that the $R^2$ of the regression line between the confined water level and deformation on overpasses were low (0.42, 0.02, 0.46, and 0.04). Figure 13c,e,g show that the monthly average deformation has a delayed response to the change of the confined water level on Sifang overpass (No.12), Pingfang overpass (NO.16), and Beipu overpass (NO.23). We then used the PDL model [45] to calculate the lag of the seasonal deformation response to the confined water level. The PDL model assumes that the impact of water-level variations on the seasonal deformation before several months should be greater than the immediate impact. In this study, the deformation at month t was used as the response variable and the confined water level at the current month and several months before (t, t − 1, . . . , t − i) were used as predictor variables. The lag months were determined when the $R^2$ reached the maximum. As can be seen in Table 5, the response of the overpass deformation to the change of the groundwater level in the subsidence area has a lag of 3–5 months on 4 overpasses. In addition, the deformation of the overpasses (Pingfang overpass (NO.16) and Beipu overpass (NO.23)) in the settlement area has a longer lag to the change of the groundwater level. Figure 14 shows the comparison of the PDL model results with the monthly deformation of the overpasses. In terms of the trend, the predicted value of the PDL model is consistent with the change trend of the deformation, and the maximum absolute error is 1.1 mm. It demonstrates that the PDL model can more accurately reflect the lagging effect existing in the deformation and groundwater-level variations.

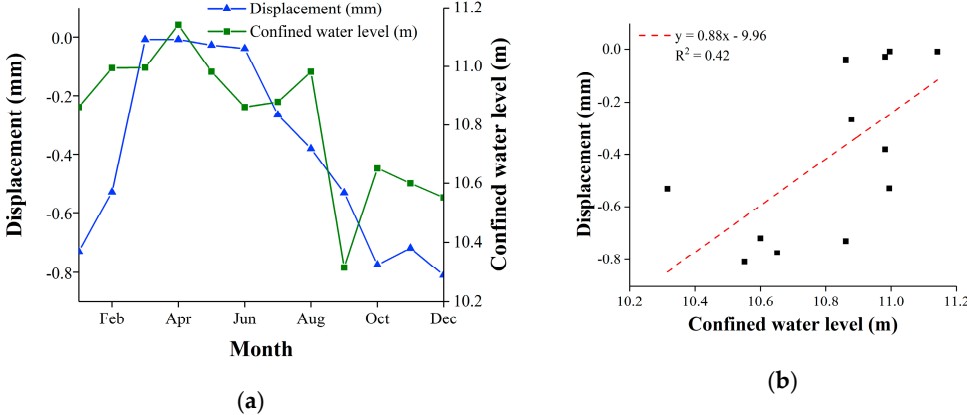

(a)

(b)

**Figure 13.** *Cont.*

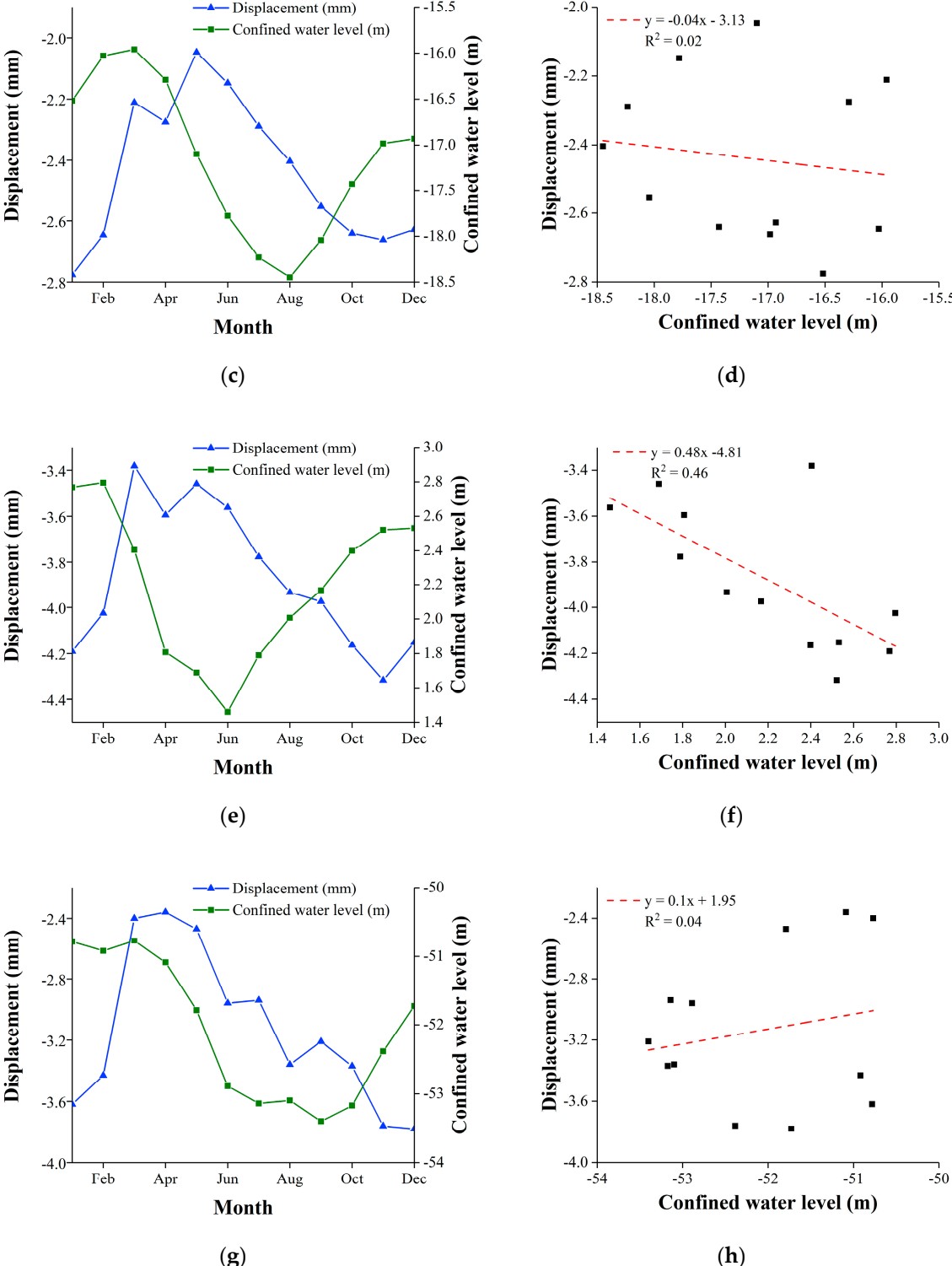

**Figure 13.** The relationship between deformation and groundwater level. The trend of monthly deformation and groundwater level on (**a**) Siyuan overpass (NO.9), (**c**) Sifang overpass (No.12), (**e**) Pingfang overpass (NO.16), and (**g**) Beipu overpass (NO.23); The correlation between monthly deformation and groundwater level on (**b**) Siyuan overpass (NO.9), (**d**) Sifang overpass (No.12), (**f**) Pingfang overpass (NO.16), and (**h**) Beipu overpass (NO.23).

**Table 5.** The delayed month of the seasonal deformation response to the confined water level on 4 overpasses.

| Overpass Number | Deformation Rate (mm/year) | Lag Period (Month) | $R^2$ | PDL Model |
|---|---|---|---|---|
| 9 | −5.11 | 3 | 0.92 | $\hat{y} = -18.215 + 0.5097x_t + 0.59x_{t-1} + 0.4517x_{t-2}$ $+ 0.0948x_{t-3}$ |
| 12 | −30.02 | 3 | 0.79 | $\hat{y} = -0.4011 - 0.1715x_t + 0.0823x_{t-1} + 0.1565x_{t-2}$ $+ 0.0.0511x_{t-3}$ |
| 16 | −47.27 | 5 | 0.75 | $\hat{y} = -4.6079 - 0.203x_t - 0.05x_{t-1} + 0.0657x_{t-2}$ $+ 1.4435x_{t-3} + 0.1857x_{t-4}$ $+ 0.1899x_{t-5}$ |
| 23 | −39.11 | 5 | 0.83 | $\hat{y} = -92.5641 + 0.0882x_t + 0.056x_{t-1} + 0.1116x_{t-2}$ $+ 0.2549x_{t-3} + 0.4860x_{t-4}$ $+ 0.8049x_{t-5}$ |

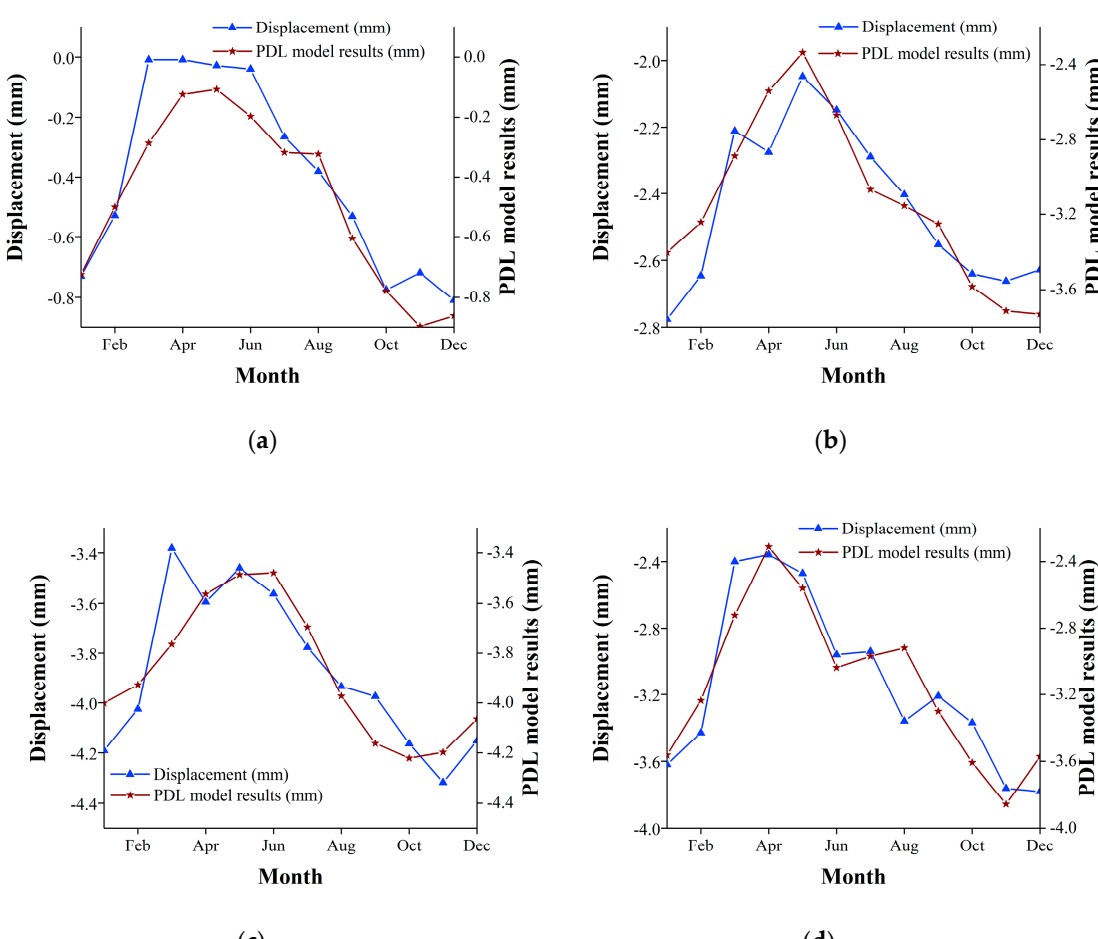

**Figure 14.** The trend of monthly deformation and PDL model results on (**a**) Siyuan overpass (NO.9), (**b**) Sifang overpass (No.12), (**c**) Pingfang overpass (NO.16), and (**d**) Beipu overpass (NO.23).

## 5.2. The Relationship with Temperature

Man-made structures, such as steel core bridges and specific buildings, may be very sensible to thermal dilation effects [5]. Warmer periods in spring and summer may lead to concrete expansion. Conversely, colder periods in autumn and winter cause shrinking and hardening of concrete [47]. The results in Section 4.2 show that the settlement of overpasses increased in autumn and winter, and slowed down in spring and summer. Although temperature variations may cause both horizontal and vertical deformation of the structures, here we only focused on the vertical deformation. In this study, monthly temperature observed at Chaoyang observation station (Figure 1) was acquired from National Meteorological Centre of China (http://data.cma.cn/). Figure 15a demonstrates the average monthly deformation rates of all seasonal PS points on the upper-level overpasses in the study area and the average monthly temperature from 2010 to 2016. It shows that the temporal trend of monthly

deformation is basically consistent with that of the monthly temperature. The monthly deformation and temperature change were high in March–June and were low in October–December. Figure 15b shows that temperature and deformation have a high correlation, with $R^2$ reaching 0.77. In addition, in both the subsidence area and non-subsidence area, the monthly vertical deformation of seasonal PS points on the upper-level overpasses showed high correlations with temperature (Figure 15d,f). This suggests that temperature may be an important factor causing seasonal deformation of the upper-level overpasses.

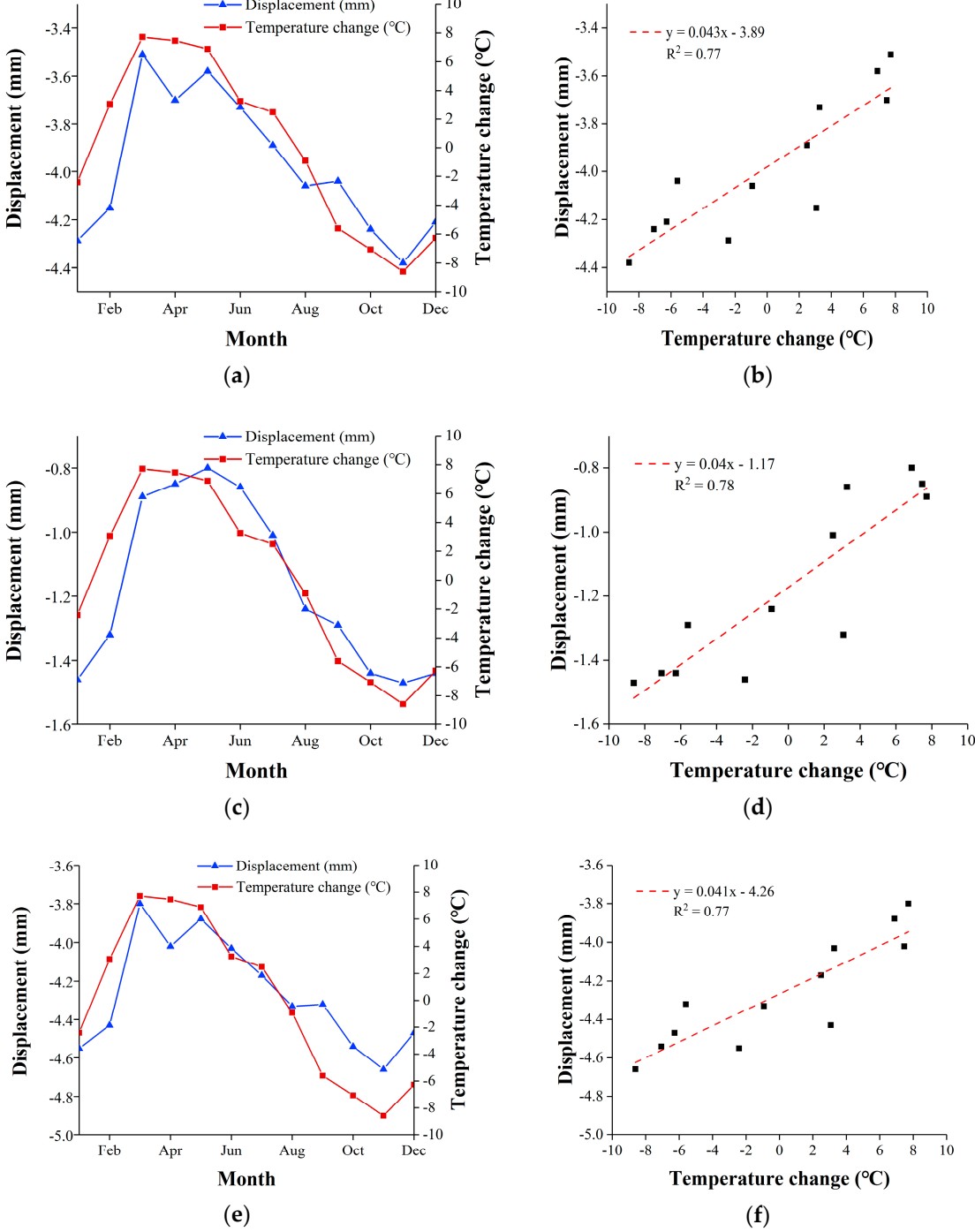

**Figure 15.** Monthly displacement and temperature in (**a**) the study area, (**c**) non-subsidence area, and (**e**) subsidence area; the scatterplots of monthly displacement and temperature on the upper-level overpasses in (**b**) the study area, (**d**) non-subsidence area, and (**f**) subsidence area.

*5.3. The Relationship with the Compressible Layer*

The groundwater drawdown in the aquifer is the inducement of land subsidence and the compressible soil can control the magnitude of land subsidence [48,49]. The compressible layers are the main contributors to land subsidence in Beijing. Zhou et al. used the gradient lifting decision tree (GBDT) model to quantitatively analyze the multiple factors of land subsidence, and found the compressible thickness and groundwater-level contribution to land subsidence exceeded 60% [46]. The first aquifer below the first compressible layer is usually considered to be the optimal water resource for daily life and industrial and agricultural production because of the shallow depth and high water yield [50]. Therefore, the first compressible layer is easily influenced by groundwater change and is the major contributor to land subsidence. From Figure 16, with the increase of the thickness of the first compressible soil layer, the seasonal characteristics of the overpasses are more obvious. According to the results obtained by Cao et al. in the experiment, the rebound deformation of two sample points in the first compressible layer caused by groundwater recharge were 0.07 and 0.112 mm, respectively (depths of 12 and 21 m, respectively) [50]. It indicated that the greater the thickness is, the greater the elastic shape variable caused by groundwater change in the first compressible layer, and the more obvious the characteristics of seasonal deformation will be. In this study, the compressible layer thickness was classified into three classes, including 70–120 m, 120–170 m, and 170–220 m. From Table 6, the number of seasonal overpasses with a thickness of 170–220 m in the compressible sediment is the largest, accounting for 100% of the total number of overpasses with the same compressible layer thickness. Therefore, the compressible thickness provides a favorable geological background for the seasonal deformation characteristics of the overpasses.

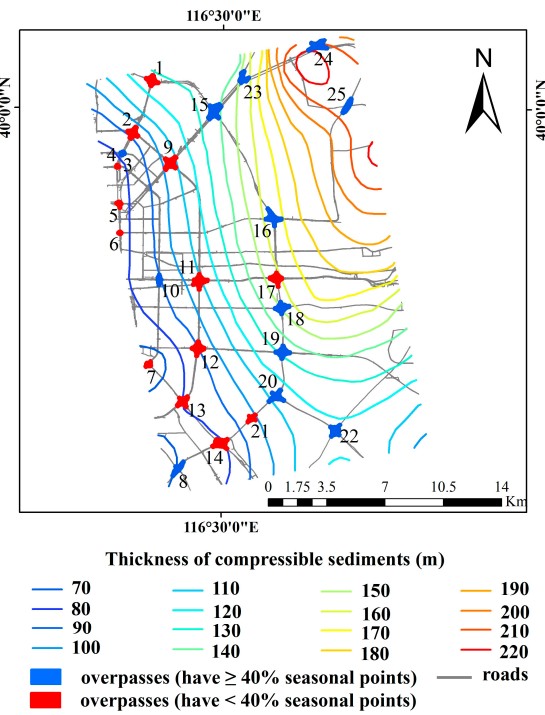

**Figure 16.** Seasonal differences of the overpasses in different thicknesses of compressible sediment.

**Table 6.** Seasonal differences of the overpasses in different thicknesses of compressible sediment.

| Thickness of Compressible Layer (m) | Number of Overpasses | Overpasses with Seasonal PS Points > 40% |
|---|---|---|
| 70–120 | 16 | 5 |
| 120–170 | 7 | 5 |
| 170–220 | 2 | 2 |

### 5.4. The Relationship with the Construction Age of Overpasses

Table 7 shows that newly constructed overpasses demonstrated more seasonality features than the old overpasses. Among the 10 overpasses constructed before 2002, only one overpass showed a seasonality characteristic, that is, less than 40% of the PS points showed a seasonal pattern. There were 15 overpasses constructed after 2002. Eleven of them showed seasonality. Particularly, 2 of 3 overpasses constructed after 2008 showed seasonality. This was consistent with previous research on building deformation. Solari et al. demonstrated the correlation between the age of the construction of buildings and the subsidence characteristics in two small urban areas in Italy [51]. Yang et al. found that newer blocks had greater spatial unevenness and temporal instability than older buildings in Beijing [25]. Therefore, the seasonal characteristics of settlement on the overpass may have a certain correlation with the construction age.

**Table 7.** Seasonal differences of the overpasses in different ages of construction.

| Construction Year | Number of Overpasses | Overpasses with Seasonal PS Points > 40% |
|:---:|:---:|:---:|
| 1980 | 1 | 0 |
| 1982 | 1 | 0 |
| 1984 | 1 | 1 |
| 1990 | 3 | 0 |
| 1993 | 1 | 0 |
| 1994 | 1 | 0 |
| 1995 | 1 | 0 |
| 1999 | 1 | 0 |
| 2002 | 3 | 2 |
| 2003 | 9 | 7 |
| 2008 | 3 | 2 |

## 6. Conclusions

In this study, we proposed DCD and DCP indices to characterize seasonal variations of deformation based on PS-InSAR time series observations of surface displacement. Taking Beijing urban area as the study area, we first used the PS-InSAR method to obtain land surface vertical displacement from 2010 to 2016 based on 55 TerraSAR-X images. For each PS pixel, DCD and DCP were calculated. The PS pixels with a DCD value greater than 0.3 were considered as those that have an obvious seasonal feature, suggesting that vertical displacement during certain months was considerable greater than that during the others. DCP represented the period within which the PS pixel had the highest displacement value (smallest settlement or greatest uplift). DCD and DCP were useful to understand the spatial distribution of seasonality on the overpass.

Our results showed that the maximum annual average settlement rate was −141.3 mm/year from 2010 to 2016, and two large settlement funnel areas were formed. Our PS-InSAR measurements agree well with levelling benchmark observations, with $R^2$ over 0.97 and RMSE less than 7.54 mm/year. The overpasses located in the subsidence area showed a more obvious seasonal pattern than those in the non-subsidence area, and the newly-constructed overpasses had more remarkable seasonal deformation than the older overpasses. The upper-level overpasses showed a more visible seasonal pattern than the lower-level highways. Settlement in autumn and winter is more serious than that in spring and summer.

Furthermore, we discussed the relationship between the seasonal deformation with the groundwater level, temperature, compressible layer, and construction age of overpasses. We found that there exists a time lag (3–5 months) between the deformation of overpasses and groundwater-level changes by using the PDL model. We determined that the seasonal deformation trend of overpasses is consistent with the temperature change trend. We also found that the thicker compressible soil thickness provides a favorable geological background. Our analysis showed that the seasonal vertical

deformation in the study area is affected by mutual interactions of multiple factors, such as the groundwater level, temperature, compressible layer, and construction age of overpasses.

**Author Contributions:** Conceptualization, X.L.; methodology, X.L. and Y.K.; software, M.L. and L.G.; validation, M.L.; formal analysis, M.L.; investigation, Y.K.; resources, X.L., H.G., and L.Z.; data curation, M.L. and L.G.; writing—original draft preparation, M.L.; writing—review and editing, Y.K. and X.L.; visualization, M.L.; supervision, X.L. and L.Z.; project administration, H.G.; funding acquisition, Y.K. and H.G. All authors have read and agreed to the published version of the manuscript.

**Funding:** This research was funded by Beijing Natural Science Foundation (grant number 5172002), National Natural Science Foundation of China (grant number 41401493), Capacity Building for Sci-Tech Innovation—Fundamental Scientific Research Funds, and by Beijing outstanding Young Scientist (grant number BJJWZYJH01201910028032), National Natural Science Foundation of China (number 41930109/ D010702).

**Acknowledgments:** We thank for the China Geological Survey (CGS) for the levelling data released to the public. We also thank to China Geological Environmental Monitoring Institute (CIGEM) for sharing groundwater level measurement data. We also thank to National Meteorological Centre of China (NMCC) for supplying temperature data. We also thank the National Aeronautics and Space Administration (NASA) for making the SRTM DEM data available. Moreover, we also thank the creators of the SARPROZ, ArcGIS and EViews software.

**Conflicts of Interest:** The authors declare no conflict of interest.

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
