# Peer review of "Detection of Seasonal Deformation of Highway Overpasses Using the PS-InSAR Technique: A Case Study in Beijing Urban Area"

_remotesensing, doi:10.3390/rs12183071_

Round 1

Reviewer 1 Report

The authors have improved the corrected manuscript, according to the suggestion of the reviewers, however, the paper still to need further improvements before to be published.  

In general, it still to lack some more analysis that shows that the seasonal, but also the general trend of the overpass is different from the rest of the surrounding area. Consider presenting DCD and DCP as tools to detect  seasonal trend on deformation 

other considerations

Figure 1: The administrative boundaries in the overpass map are not necessary 

Figure 8: There is no large difference between a) 45% and b) 50%  Maybe it is  better to make one figure in which overpasses are characterized by the percentage of seasonal PS (e.g., 25 %, 50 %, 75 %,..) 

Figure 9:   A map where is shows for all ps dataset the distribution of seasonal and not seasonal PS should be useful, to understand how the single overpass are related to their surrounding areas.  

Figures 13 and 14: please use the same deformation range   for all figures (in figure 14 c is 0.8 mm while for in figures e is 2.4 mm, and this imply that deformation ha stronger variability and from  the plot  is not evident) 

conclusions still need to be improved. 

Author Response

Dear Reviewer:

Thank you for your letter and for the reviewers’ constructive comments concerning our manuscript entitled “Detection of Seasonal Deformation of Highway Overpasses Using PS-InSAR Technique: a case study in Beijing urban area” (ID: 890977). We have made careful modifications on the original manuscript according to the reviewers' comments. Revised portion are marked in Green in the paper. The main corrections in the article and the responses to the comments are as flowing:

Responds to your comments:

  1. Response to comment: (Figure 1: The administrative boundaries in the overpass map are not necessary.)

Response: Thanks for your advice. We have modified the original Figure 1. In the new Figure 1, we have removed the administrative boundaries.

  1. Response to comment: (Figure 8: There is no large difference between a) 45% and b) 50%. Maybe it is better to make one figure in which overpasses are characterized by the percentage of seasonal PS (e.g., 25 %, 50 %, 75 %,..))

Response: Thanks for your advice. According to your suggestion, we have modified Figure 8 and Table 3, to show the spatial difference of the seasonal distribution of overpasses. We have further revised the texts in Line 320-323 to explain the figure.

  1. Response to comment: (Figure 9: A map where is shows for all PS dataset the distribution of seasonal and not seasonal PS should be useful, to understand how the single overpass are related to their surrounding areas.)

Response: Thanks for your suggestion. Because of the large image coverage area and the small number of overpasses, the distribution difference of all seasonal and non-seasonal PS points is not obvious. We have modified the original Figure 9. The new Figure 9 shows the distribution of seasonal and non-seasonal PS points in four typical overpasses and surrounding areas. We have also extended the corresponding descriptions in Line 338-340.

  1. Response to comment:(Figures 13 and 14: please use the same deformation range for all figures (in figure 14 c is 0.8 mm while for in figures e is 2.4 mm, and this imply that deformation ha stronger variability and from the plot is not evident))

Response: Thanks for your advice. As can be seen from the left coordinates of Figure 13 and Figure 15(Figure 14, originally), the monthly settlement ranges of the overpasses located in different areas are quite different. Taking Figure 15 (a) Siyuan Overpass (NO.9),) and (e) Pingfang Overpass as examples, the two overpasses are located in the non-settlement area and settlement area respectively, with the maximum monthly deformation reaching -0.8mm and -4.3mm respectively, and the annual deformation being -4.8mm and -46.53mm respectively, which are of great difference. If the graph is shown in the same deformation range, the fluctuation of deformation appears to be particularly small, which is hard to reflect the seasonal character of deformation. Therefore, we did not change it.

  1. Response to comment: (conclusions still need to be improved.)

Response: Thanks for your advice. We have modified the conclusion in Line 513-518. We have added detailed description of the Discussion Section.

Thank you very much for your work concerning our paper. We hope the new manuscript will meet with approval.

Sincerely yours,

Ms. Mingyuan Lyu

Reviewer 2 Report

This manuscript describes work where PS-InSAR is used to determine seasonal deformation in highway overpasses in and around Beijing. The work introduces new metrics for assessing seasonality: deformation concentration degree (DCD) and deformation concentration period (DCP). The work compares the PS-InSAR results with leveling results, which are considered reliable, to establish the trustworthiness of PS-InSAR. This is used to assess overpasses based on the degree to which deformation for PS pixels on each overpass is seasonal. The work then identifies factors that correlate with seasonal deformation, which include overpass level.

I found the results exciting, and I suspect that they are correct. However, there are a number of issues that I feel need to be addressed before publication. Listed from most important to least important they are: 

  1. DCD and DCP appear to have undesirable behavior in regards to sign changes. Consider the hypothetical case that there is an uplift in July of 1 unit, and no deformation at any other month. DCD will be one, and DCP will point to July. However, if there is 1 unit of subsidence in July, and no deformation at any other month, then adding the absolute value of the minimum deformation (Note that the language describing Eq. 3 differs from what Eq. 3 actually is) will indicate an uplift of 1 at every month other than July. If I understand, this will give a DCD of 1/11 and a DCP of January. More to the point, based on my understanding of DCD and DCP, I disagree with the claim, “This processing does not change the trend of temporal distribution of deformation on each pixel.”

    There are a lot of ways to assess seasonality. As a simple approach, one could remove a linear trend for each year, and look at the yearly peak-to-peak difference of the average of the monthly residuals. Alternatively, there are established Python packages that will separate seasonal trends from longer term trends.

    I believe that the important result in this manuscript is that there is seasonality in deformation of overpasses, I think the question of whether DCD and DCP are the right metric seems beside the point.

    As an aside, the spread in values for each r_i would be useful.
  2. InSAR is always a relative measurement. One can only make measurements of one location relative to another location. In reading this manuscript I was unable to determine what was used as a reference point. This is particularly important because the distance from the reference point to the test point will indicate the degree to which APS will play a role. If a reference point is within a km of an overpass, then APS contributions will be minimal. On the other hand, some of the seasonal effects such as groundwater level, might influence the reference point as well. For the reader to assess this work and its value in other settings, the choice of reference point(s) needs to be made clear.
  3. Leveling is used as a test of InSAR accuracy, but the results are summarized at too high a level. The manuscript needs more description for Figure 6. In particular, what leveling measurement is being made here? Was this a measurement of one point relative to another point? If yes, how far apart were these points? How are 6b and 6c different? The trends in 6b and 6c are remarkably linear, so much so that one cannot see seasonality. Indeed if there is seasonality, it looks to be about the same magnitude as the difference between the leveling measurement and the InSAR measurement. It might be better to subtract off the linear trend before plotting this. 
  4. The left hand plots of figures 13 and 14 used a temporal smoothing process that needs better description and justification. The plots on the right hand side of figures 13 and 14 indicate distinct points, presumably these came from the smoothed data used on the left, but one cannot determine which points were used. In particular, I cannot reconcile Figures 13 e and f. Additionally, I would think that the authors would want to produce plots of displacement vs. the output of the PDL model.
  5. At line 169 Eq. 1 is described as the main formula of the PS-InSAR method. It’s more a mathematical description of contributions to phase for a wrapped interferogram.
  6. Table 1 should include the mode in which TerraSAR-X was operating.
  7. Some of the text was red. 
  8. In Figure 9 seasonal and non-seasonal pixels, even on what appears to be the same level of structure, are intermingled. Is seasonal deformation such a localized phenomenon, or are we seeing some variability to the method? Some discussion would be useful. I believe the right-most plot in Figure 9 should be Figure 9d, it’s currently marked as Figure 9b.

Author Response

Dear Reviewer:

Thank you for your letter and for the reviewers’ constructive comments concerning our manuscript entitled “Detection of Seasonal Deformation of Highway Overpasses Using PS-InSAR Technique: a case study in Beijing urban area” (ID: 890977). We have made careful modifications on the original manuscript according to the reviewers' comments. Revised portion are marked in Green in the paper. The main corrections in the article and the responses to the comments are as flowing:

Responds to your comments:

  1. Response to comment: (DCD and DCP appear to have undesirable behavior in regards to sign changes. Consider the hypothetical case that there is an uplift in July of 1 unit, and no deformation at any other month. DCD will be one, and DCP will point to July. However, if there is 1 unit of subsidence in July, and no deformation at any other month, then adding the absolute value of the minimum deformation (Note that the language describing Eq. 3 differs from what Eq. 3 actually is) will indicate an uplift of 1 at every month other than July. If I understand, this will give a DCD of 1/11 and a DCP of January. More to the point, based on my understanding of DCD and DCP, I disagree with the claim, “This processing does not change the trend of temporal distribution of deformation on each pixel.”

There are a lot of ways to assess seasonality. As a simple approach, one could remove a linear trend for each year, and look at the yearly peak-to-peak difference of the average of the monthly residuals. Alternatively, there are established Python packages that will separate seasonal trends from longer term trends.

I believe that the important result in this manuscript is that there is seasonality in deformation of overpasses, I think the question of whether DCD and DCP are the right metric seems beside the point.

As an aside, the spread in values for each r_i would be useful.)

Response: First of all, Thanks very much for your reminder. We have changed the eq.3 “ ” to “ ”.

This method is more suitable for finding the concentration degree and the period of the peak (the trend of first increasing and then decreasing in deformation time series). As shown in some Figures (Figure 7, Figure 12-15), the peak of deformation occurs in spring, and the valley of deformation usually occurs in autumn. Based on this study and previous studies (Zhang, et al. 2014; Zhu, et al. 2013), it is proved that the temporal trend of ground and overpass deformation in Beijing is the situation discussed in this study.

The Python packages you mentioned are suitable for long time series data that are periodic and are frequently used for time series forecasting. Another type of methods was to use indices to measure distribution and concentration of time series variables within a certain period, and have been widely used to analyse intra-annual pattern of climatological or hydrological variables such as precipitation and streamflow. The composite vector can better reflect the concentration, barycentre date of precipitation. Therefore, we use DCD and DCP to characterize seasonal deformation of overpasses. In Line 95-105, we have added the description of the models and indices used in the previous studies of seasonal characteristics.

In the Results and Discussion Section, we have described the distribution trend of each monthly deformation on the overpasses. In future studies, we will conduct a more comprehensive analysis based on your suggestions.

References

Zhang, Y.; Gong, H.; Gu, Z.; Wang, R.; Li, X.; Zhao, W. Characterization of land subsidence induced by groundwater withdrawals in the plain of Beijing city, China. Hydrogeol. J. 2014, 22, 397-409.

Zhu, L.; Gong, H.; Li, X.; Wang, R.; Chen, B.; Dai, Z.; Teatini, P. Land subsidence due to groundwater withdrawal in the northern Beijing plain, China. Eng. Geol. 2015, 193, 243-55.

  1. Response to comment: (InSAR is always a relative measurement. One can only make measurements of one location relative to another location. In reading this manuscript I was unable to determine what was used as a reference point. This is particularly important because the distance from the reference point to the test point will indicate the degree to which APS will play a role. If a reference point is within a km of an overpass, then APS contributions will be minimal. On the other hand, some of the seasonal effects such as groundwater level, might influence the reference point as well. For the reader to assess this work and its value in other settings, the choice of reference point(s) needs to be made clear.

Response: We agree that reference points play an important role in data processing. The criterion for selecting reference points is to find the region where the phases of deformation and height are at 0 based on previous research. Secondly, according to high coherence and stable reflection characteristics, we select the optimal reference point in the found region. In the manuscript, we show the location of the reference points in the new Figure 1. The details of reference points are described in Line 203-204.

  1. 3. Response to comment:( Leveling is used as a test of InSAR accuracy, but the results are summarized at too high a level. The manuscript needs more description for Figure 6. In particular, what leveling measurement is being made here? Was this a measurement of one point relative to another point? If yes, how far apart were these points? How are 6b and 6c different? The trends in 6b and 6c are remarkably linear, so much so that one cannot see seasonality. Indeed if there is seasonality, it looks to be about the same magnitude as the difference between the leveling measurement and the InSAR measurement. It might be better to subtract off the linear trend before plotting this.)

Response: Thanks for your advice. The level monitoring data used in this study are the monitoring results of relevant research departments. The monitoring method is as follows: a large number of levelling points are set up, and a levelling monitoring network is formed, height differences are measured between established benchmarks. Unfortunately, we only got 13 levelling points data (Figure 1). For each of the levelling points, the closest PS pixels were found and compared. The level point monitoring results only have one value per year, namely the annual settlement, which cannot reflect the inter-annual change (i.e. seasonality). Therefore, we used the annual average displacement rate and cumulative displacement to assess the accuracy.

The Figure 6 (b, c) are the results of the long-time monitoring of the two PS points closest to the levelling points, which are the cumulative settlement in the vertical direction. The two PS points are located on the non-overpass buildings (Figure 1). After calculation, the DCD of these two PS points are 0.23 and 0.10, respectively, which belongs to “non-seasonal” PS point. These two points are located in the Chaoyang Jinzhan (CJ) settlement funnel area. In my previous research and historical studies, it can be seen that the seasonal fluctuations of the deformation in the settlement funnel area are weak.

In addition, we have added more description for Figure 6 (Line283 -285).

  1. Response to comment:( The left hand plots of figures 13 and 14 used a temporal smoothing process that needs better description and justification. The plots on the right hand side of figures 13 and 14 indicate distinct points, presumably these came from the smoothed data used on the left, but one cannot determine which points were used. In particular, I cannot reconcile Figures 13 e and f.? Additionally, I would think that the authors would want to produce plots of displacement vs. the output of the PDL model. )

Response: We are sorry for the confusion here. We modified the original Figure 13 and original Figure 14(new Figure 15), changed the curve to line graph connected by points and highlighted the scattered points, making it easier for the reader to find which points were used. The description for new Figure 15 is extended in Line 449-450.

We have added a new figure (Figure 14) to show the relationship with deformation and PDL model results. The description for new Figure 14 is described in Line 423-427.

  1. Response to comment: (At line 169 Eq. 1 is described as the main formula of the PS-InSAR method. It’s more a mathematical description of contributions to phase for a wrapped interferogram.)

Response: Thanks for your advice. In Line 187-194, we have added more mathematical description of contributions to phase for a wrapped interferogram.

  1. Response to comment: (Table 1 should include the mode in which TerraSAR-X was operating.)

Response: Thanks for your advice. In Line 141-142, we have added the operational mode of TerraSAR-X.

  1. Response to comment: (Some of the text was red.)

Response: This article had been submitted once before, so this is a resubmitted vision with red revisions.

  1. Response to comment: (In Figure 9 seasonal and non-seasonal pixels, even on what appears to be the same level of structure, are intermingled. Is seasonal deformation such a localized phenomenon, or are we seeing some variability to the method? Some discussion would be useful. I believe the right-most plot in Figure 9 should be Figure 9d, it’s currently marked as Figure 9b.)

Response: Thanks for your reminder. The error of “(b)” has been changed to “(d)”. We also thanks for your suggestion. We have modified the original Figure 9. The new Figure 9 shows the distribution of seasonal and non-seasonal PS points in the overpass and surrounding areas. We have also added the corresponding descriptions in Line 338-340. In addition, we divided the overpass into upper-level overpass and lower-level highway, and it can be seen that seasonal PS points are more concentrated in the upper-level overpass (Figure 10).

Thank you very much for your work concerning our paper. We hope the new manuscript will meet with approval.

Sincerely yours,

Ms. Mingyuan Lyu

Reviewer 3 Report

This paper proposes a method for detection of seasonal deformation of highway overpasses using the integration of persistent scatterers Interferometric Synthetic Aperture Radar (PS-InSAR) techniques. deformation concentration degree (DCD) and deformation concentration period (DCP) indices. The presented methodology is generally clear but it does not reflect the scientific soundness of the proposed method. Therefore, The authors should present their methods more clearly.

As an example the introduction of the manuscript should provide a wider description of the existing methodologies. In this way it would be easier for the reader to understand the advancements provided in the paper and proposed methodology.

 It is very hard at this stage to contextualize the manuscript and the proposed advancements. I recommend widening the introduction and provide reference to important literature in the field such as the one below. I am willing to further review the rest of the paper after my comments are addressed.

Milillo, Pietro, et al. "Pre-collapse space geodetic observations of critical infrastructure: the Morandi bridge, Genoa, Italy." Remote Sensing 11.12 (2019): 1403.

Giardina, Giorgia, et al. "Evaluation of InSAR monitoring data for post‐tunnelling settlement damage assessment." Structural Control and Health Monitoring 26.2 (2019): e2285.

Ozden, Abdulkadir, et al. "Evaluation of Synthetic Aperture Radar satellite remote sensing for pavement and infrastructure monitoring." Procedia Engineering 145 (2016): 752-759.

Zhao, J., Wu, J., Ding, X., & Wang, M. (2017). Elevation extraction and deformation monitoring by multitemporal InSAR of Lupu Bridge in Shanghai. Remote Sensing, 9(9), 897.

Author Response

Dear Reviewer:

Thank you for your comments concerning our manuscript entitled “Detection of Seasonal Deformation of Highway Overpasses Using PS-InSAR Technique: a case study in Beijing urban area” (ID: 890977). Those comments are all valuable and very helpful for revising and improving our paper, as well as the important guiding significance to our researches. We have studied comments carefully and have made correction which we hope to meet with your approval. Revised portion are marked in Green in the paper. The main corrections in the article and the responses to the comments are as flowing:

Responds to your comments:

  1. Response to comment: (This paper proposes a method for detection of seasonal deformation of highway overpasses using the integration of persistent scatterers Interferometric Synthetic Aperture Radar (PS-InSAR) techniques. deformation concentration degree (DCD) and deformation concentration period (DCP) indices. The presented methodology is generally clear but it does not reflect the scientific soundness of the proposed method. Therefore, The authors should present their methods more clearly.

Response: Thanks for your advice. In the revised manuscript, we have improved Section 1 “Introduction” section and Section 3.2 “Detection of seasonal deformation using DCD and DCP” in order to justify the scientific soundness of the methods. In Section 1, we have added some literatures and showed that a few studies have found seasonal variations of deformation on viaduct and bridges by analyzing time-series several PS points but there has been no study for automatic detection of seasonality of deformation. Detection and understand deformation seasonality could help us better understanding the mechanism of urban infrastructure deformation. In Section 3.2, we made additional description of DCD and DCP, and justify the suitability of the method for seasonal deformation detection. Please see Line 43-119 and Line 212-253.

  1. Response to comment: (As an example the introduction of the manuscript should provide a wider description of the existing methodologies. In this way it would be easier for the reader to understand the advancements provided in the paper and proposed methodology. It is very hard at this stage to contextualize the manuscript and the proposed advancements. I recommend widening the introduction and provide reference to important literature in the field such as the one below. I am willing to further review the rest of the paper after my comments are addressed.

Milillo, Pietro, et al. "Pre-collapse space geodetic observations of critical infrastructure: the Morandi bridge, Genoa, Italy." Remote Sensing 11.12 (2019): 1403.

Giardina, Giorgia, et al. "Evaluation of InSAR monitoring data for post‐tunnelling settlement damage assessment." Structural Control and Health Monitoring 26.2 (2019): e2285.

Ozden, Abdulkadir, et al. "Evaluation of Synthetic Aperture Radar satellite remote sensing for pavement and infrastructure monitoring." Procedia Engineering 145 (2016): 752-759.

Zhao, J., Wu, J., Ding, X., & Wang, M. (2017). Elevation extraction and deformation monitoring by multitemporal InSAR of Lupu Bridge in Shanghai. Remote Sensing, 9(9), 897.)

Response: Thank you very much for your suggestions. We have added your recommended references in the Introduction Section (Line 53-54, Line 60-61 and Line 80-83). Milillo et al. revealed that the bridge was undergoing an increased magnitude of deformations over time prior to its collapse. Zhao et al. found seasonal deformation pattern for several PS points in Lupu Bridge in Shanghai and analysed the relationship with temperature. With the development of SAR technology, SAR-based monitoring has become valuable for monitoring the deformation of infrastructure elements such as bridge displacement, roadway surface deformation, structural damage assessment, etc. It also has the potential of structural damage assessment.

In addition, we have added the description of the existing methodologies in previous studies (Line 95-105). It is shown that DCD and DCP are more suitable to find the seasonal characteristics of deformation.

Thank you very much for your work concerning our paper. We hope the new manuscript will meet with approval.

Sincerely yours,

Ms. Mingyuan Lyu

Round 2

Reviewer 1 Report

After this revision 

The paper can be accepted for publication.

Please only provide to use another color for dark red  label in figure 1

Author Response

Dear Reviewer:

Thank you for your letter and for the reviewers’ constructive comments concerning our manuscript entitled “Detection of Seasonal Deformation of Highway Overpasses Using PS-InSAR Technique: a case study in Beijing urban area” (ID: 890977). We have made modifications on the original manuscript according to the reviewers' comments. Revised portion are marked in Blue in the paper. The main corrections in the article and the responses to the comments are as flowing:

Response to your comment: (Please only provide to use another color for dark red label in Figure 1)

Response: Thanks for your advice. We have modified the original Figure 1. In the new Figure 1, we have changed the color of some elements, such as the Overpass, Levelling benchmark (2010-2013) and the Name of district.

We appreciate for the reviewer’ s and editors’ warm work earnestly. Once again, thank you very much for your comments and suggestions.

Sincerely yours,

Ms. Mingyuan Lyu

Reviewer 2 Report

The most recent version of the manuscript includes most of the changes that I wanted to see, and I think this most recent version is worthy of publication. 

In my opinion the strength of this paper is the application of InSAR to an important issue that has the potential to affect people's lives. Further the analysis struck me as thorough -- I was impressed and gratified that, e.g., the authors separated analysis by levels of an overpass; that exciting to read about! 

I also think that DCD and DCP are flawed metrics, but the authors seem committed to holding onto them. As best I can tell, CD and CP work well for unsigned temporal data, but the analogues presented here have a failure mode for signed temporal data, which I tried to explain in my earlier review. This isn't worth holding up publication, but I encourage the authors to consider other temporal analysis methods. 

Author Response

Dear Reviewer:

Thank you for your letter and for the reviewers’ constructive comments concerning our manuscript entitled “Detection of Seasonal Deformation of Highway Overpasses Using PS-InSAR Technique: a case study in Beijing urban area” (ID: 890977). We have made careful modifications on the original manuscript according to the reviewers' comments. Revised portion are marked in Blue in the paper.

Response to comment: (The most recent version of the manuscript includes most of the changes that I wanted to see, and I think this most recent version is worthy of publication.

In my opinion the strength of this paper is the application of InSAR to an important issue that has the potential to affect people's lives. Further the analysis struck me as thorough -- I was impressed and gratified that, e.g., the authors separated analysis by levels of an overpass; that exciting to read about!

I also think that DCD and DCP are flawed metrics, but the authors seem committed to holding onto them. As best I can tell, CD and CP work well for unsigned temporal data, but the analogues presented here have a failure mode for signed temporal data, which I tried to explain in my earlier review. This isn't worth holding up publication, but I encourage the authors to consider other temporal analysis methods.)

Response: Thank you very much for your work concerning our paper. According to your suggestion, we will try more methods for temporal analysis in future research.

We appreciate for the reviewer’ s and editors’ warm work earnestly. Once again, thank you very much for your comments and suggestions.

Sincerely yours,

Ms. Mingyuan Lyu

This manuscript is a resubmission of an earlier submission. The following is a list of the peer review reports and author responses from that submission.

Round 1

Reviewer 1 Report

Major comments:

  1. Section 2.2: Please provide the dataset acquisition parameters such as flight mode, incidence angle, etc.
  2. Section 3.1: The interferometric processing parameters were not given sufficient details such as baseline information, etc. The author have mentioned the input SAR data was acquired at 3m, could you provide the output spatial resolution of the interferometric products. Was there any multi-looking applied in the interferogram generation. And for the deformation analysis, which product was used? PS-InSAR results in radar co-ordinates or geocoded coordinates.
  3. Did the authors apply any atmospheric corrections to the PS-InSAR results?
  4. Line 231: why the east displacement is higher than the west displacement, please comment on this.
  5. Figure 5: The authors should mention the location of PS point chosen for this comparison, there are several leveling points in the study area. Could you please mention whether the displacement trend in 5b and 5c is LOS or Vertical. And, the displacement trend does not have any seasonal displacement in Fig 5b and 5c. How do you explain this characteristic?
  6. This paper aims to analyze the deformation on the infrastructures (overpasses), but most of the results only analyze the temporal deformation trend obtained from InSAR displacement results. Except for Fig. 4 there is no spatial distribution of displacement over the overpasses. The spatial distribution is equally important to understand the seasonal deformation on the infrastructures.
  7. The author should describe the acquisition information and quality of the auxiliary data such as temperature dataset, groundwater level data
  8. I recommend you to analyze the spatial and temporal distribution of PS-InSAR results with temperature, groundwater in detail for at least two overpasses.

Minor comments:

  1. Line 43: the citation type [1] and [2] are incorrect.
  2. Figure 4: Is this LOS displacement rate or vertical displacement rate?
  3. Please provide describe the acquisition of temperature dataset
  4. The manuscript needs language editing

Reviewer 2 Report

In this manuscript, seasonal deformation of highway overpasses in the Beijing urban area, is detected, applying the persistent scatterers Interferometric Synthetic Aperture Radar (PS-InSAR) technique, implemented in the SARPROZ software, and introducing two indices (DCD and DCP), used to characterize seasonal variations of the deformation. The analysis showed that the seasonal characteristics of deformation in the study area are affected by temperature, groundwater level, compressible layer and construction age of overpasses.

Unfortunately, the only novel contribution of the manuscript is the introduction of deformation concentration degree (DCD) and deformation concentration period (DCP) indices. CD and CP indeces were first proposed by Zhang et al. (2003) to evaluate the seasonal characteristics of precipitation [29], and here they have been adapted to describe seasonal characteristics of deformation. Since the aim of the journal is to publish novel / improved methods / approaches and / or algorithms of remote sensing to benefit the community, in my opinion the manuscript is not suitable for publication.

Reviewer 3 Report

Dear authors, 

This paper presents an interesting wok on the detection of seasonal deformation that affected bridges in subsidence areas using InSAR data. The paper needs several improvements to be published.

Here the points that need improvements

  1. The abstract should be shortened 
  2. Revise that all the citations the text are expressed according to Remote Sensing guidelines.
  3. Add some  more references about the  role of the compressible layer and subsidence  made with InSAR (e.g . Béjar-Pizarro  et al., 2015)
  4. Add some  more references about the  role of thermal dilation  and subsidence  made with InSAR (e.g. Crosetto et al., 2015) 
  5. Fig. 5:  Show on the map where are located the levelling benchmark 
  6. The seasonal variation related to the subsidence may be stronger than thermal oscillation. Maybe it's better to study these two effects separately. Use overpass located in stable areas to study thermal component and use the overpass in strong subsidence area to study groundwater level.
  7. Correct some type like PCP=140 in figure 3; 
  8. Consider that the thermal dilation has also horizontal component and its incorrect use subsidence and uplift as you used in the conclusions.
  9. The conclusion needs improvement and the results better explained avoid the "more obvious"  
  10. Map in which PS are classified by DCD and DCP  should be useful to understand the spatial distribution of seasonality with an overall view  that surrounds the overpass 
  11. Figure 8 is hard to  understand 
  12. Figure 7 needs a better caption with more explanation.
  13. The old overpasses were built in areas less affected by subsidence it is important to consider the mutual interaction of all the factors that you used